# Towards Smart Blades for Vertical Axis Wind Turbines: Different Airfoil Shapes and Tip Speed Ratios

M. Rasoul Tirandaz[1], Abdolrahim Rezaeiha[2,3], Daniel Micallef[1]

[1] University of Malta, Department of Environmental Design, Msida, MSD2080, Malta
[2] KU Leuven, Leuven, Belgium
[3] Eindhoven University of Technology, Eindhoven, The Netherlands

*Correspondence to*: M. Rasoul Tirandaz (msctirandaz@gmail.com)

**Abstract.** Future wind turbines will benefit from state-of-the-art technologies that allow them to not only operate efficiently in any environmental condition, but also to maximize the power output and cut the cost of energy production. Smart technology, based on morphing blades, is one of the promising tools that could make this possible. The present study serves as a first step towards designing morphing blades as functions of azimuthal angle and tip speed ratio for vertical axis wind turbines. The focus of this work is on individual/combined quasi-static analysis of three airfoil shape-defining parameters, namely the maximum thickness $t/c$ and its chordwise position $xt/c$ as well as the leading-edge radius index $I$. A total of 126 airfoils are generated for a single-blade H-type darrieus turbine with a fixed blade/spoke connection point at $c/2$. The analysis is based on 630 high-fidelity transient 2D CFD simulations, previously validated with experiments. The results show that with increasing tip speed ratio, the optimal maximum thickness decreases from 24%c to 10%c, its chordwise position shifts from 35%c to 22.5%c, while the corresponding leading-edge radius index remains at 4.5. The results show an average relative improvement of 0.46, and an average increase of nearly 0.06 in $C_P$ for all the values of tip speed ratio.

**Keywords.** Smart rotor design; Morphing airfoil; Shape adaptation; Computational fluid dynamics (CFD); Floating offshore wind turbine (FOWT).

## Nomenclature

| | | | |
|---|---|---|---|
| $\alpha$ | Angle of attack [°] | $k$ | Reduced frequency, $\Omega c/2V_{ref} \approx c/2R$ [-] |
| $\alpha_{ss}$ | Static stall angle [°] | $L$ | Lift [N] |
| $\theta$ | Azimuth angle [°] | $M$ | Turbine moment [Nm] |
| $\lambda$ | Tip speed ratio, $R\Omega/U_\infty$ [-] | $n$ | Number of blades [-] |
| $\nu$ | Kinematic viscosity of air [m$^2$/s] | $P$ | Turbine output power [W] |
| $\sigma$ | Solidity, $nc/d$ [-] | $q$ | Dynamic pressure [Pa] |
| $\Omega$ | Turbine rotational speed [rad/s] | $R$ | Turbine radius [m] |
| $A$ | Turbine swept area, $h.d$ [m$^2$] | $Re_c$ | Chord-based Reynolds number, $cU_\infty\sqrt{1+\lambda^2}/\nu$ [-] |
| $c$ | Airfoil chord length [m] | $r_{LE}$ | Airfoil leading-edge radius [%c] |
| $C_d$ | Drag coefficient, $D/qA$ [-] | $T$ | Turbine thrust force [N] |
| $C_f$ | Skin friction coefficient, $D/qA$ [-] | $t/c$ | Airfoil relative maximum thickness [%] |
| $C_l$ | Lift coefficient, $L/qA$ [-] | $U_\infty$ | Freestream velocity [m/s] |
| $C_m$ | Moment coefficient, $M/(qAR)$ [-] | $U$ | Instantaneous streamwise velocity [m/s] |
| $C_P$ | Turbine power coefficient, $P/(qAU_\infty)$ [-] | $V$ | Instantaneous lateral velocity [m/s] |
| $C_T$ | Turbine thrust coefficient, $T/(qA)$ [-] | $V_{tan,n}$ | Dimensionless instantaneous tangential velocity, $(u\cos(\theta)+v\sin(\theta))/U_\infty$ [-] |
| $D$ | Drag [N] | $V_{rel}$ | Relative velocity [m/s] |
| $h$ | Turbine height [m] | $xt/c$ | Dimensionless chordwise-position of airfoil maximum thickness [%] |
| $I$ | Airfoil leading-edge radius index [-] | $TI$ | Turbulence intensity [%] |

## 1. Introduction

### 1.1 State of the art

Morphing technology has the potential to improve the performance of flying bodies by adapting their shape to different operational conditions. This can result in improved aerodynamic efficiency and the release of unwanted stresses (Debiasi et al., 2011; Wang et al., 2014). Nature has given birds the capability of continuous morphing to generate enough lift for various flight maneuvers. These bio-inspirational sources served as models for possible morphing vehicles and provided the pioneering researchers with a new method of improving aerodynamic efficiency (Wlezien et al., 1998). However, because of the technological limitations of the day, it was not possible to reach the level of smooth shape-changing capabilities as seen in birds (Barbarino et al., 2011). This led to the development of shape-changing by using ailerons, slats, flaps or variable sweep (Debiasi et al., 2011). Nowadays, advances in smart technologies have enabled such needs to be satisfied. Wing morphing is used in the aerospace industry to improve the aerodynamic efficiency and adaptability of aircraft (Ajaj et al., 2021; Yan et al., 2019), helicopters (Riemenschneider et al., 2019; Sal, 2020), micro air vehicles (Siddall et al., 2017) and unmanned air vehicles (Mir et al., 2018; Thangeswaran et al., 2019).

The blades of a wind turbine operate at relatively low wind speeds with a low level of risk. Nevertheless, morphing technology can still be of benefit for wind turbine purposes without the challenges that must be overcome in aerospace applications (e. g., additional flight control system and law to handle the complex and large-scale changes in aerodynamic surfaces at both low-speed and high-speed flight conditions) (Beyene and Peffley, 2007). The impacts of morphing blades have been extensively studied for horizontal axis wind turbines (HAWTs). For example, the effects of morphed trailing edge was studied by (Daynes and Weaver, 2012); in another work, morphing twist was found to reduce the fatigue life of turbine blades (Lachenal et al., 2013); in a work by (Macphee and Beyene, 2015) morphing blade pitch was discovered to improve the performance of HAWTs; effects of morphed trailing edge flap on the aerodynamic load control was investigated by (Zhuang et al., 2020).

The angle of attack $\alpha$ of a vertical axis wind turbine (VAWT) blade varies periodically between positive and negative values. Through this quasi-sinusoidal variation, the angle of attack $\alpha$ often exceeds the static stall angle, $\alpha_{ss}$, and the blade undergoes unsteady separation, resulting in the occurrence of dynamic stall and hysteresis effects on aerodynamic loads (Hand et al., 2017; Mulleners and Raffel, 2012; Rezaeiha et al., 2019a). When a turbine is operating at low $\lambda$, it benefits from the early stages of the dynamic stall; that is, the performance of the blade increases due to an overshoot in lift coefficient $C_l$; however, the overall power output is affected negatively by the consequential sudden drop in $C_l$ (E.Amet et al., 2009; Tirandaz and Rezaeiha, 2021). This complex aerodynamics makes the development of a single optimal airfoil for VAWTs a challenging process.

To this date, the performance of VAWTs, which very often use airfoils used in the helicopter industry (Rezaeiha et al., 2020b; Sahebzadeh et al., 2020), has been studied for airfoil parameters such as thickness-to-chord ratio $t/c$ and camber $C$ as proposed in (Song et al., 2020; Mazarbhuiya et al., 2020; Nguyen and Tran, 2015; Jain and Saha, 2020; Bianchini et al., 2015). More recently, a few studies have been conducted to improve VAWTs performance via optimizing the airfoil shape-defining parameters (e.g., maximum thickness $t/c$, chordwise position of maximum thickness $xt/c$, leading edge radius $r_{LE}$, and camber $C$) (Bedon et al., 2016; Ma et al., 2018; Ismail and Vijayaraghavan, 2015). Briefly summarized, these studies reveal that the airfoil shape strongly influences the torque characteristics and pressure distribution of the rotor, the type of stall mechanism, the aerodynamic load coefficients, namely lift and drag coefficients ($C_l$ and $C_d$), the self-starting capability, and the power coefficient of VAWTs. However, the majority of these studies, which include a few numbers of test cases, have addressed the impacts of a single parameter and keeping the others fixed. This is while, it has been shown that the airfoil shape-defining parameters have combined impacts on VAWT performance (Tirandaz and Rezaeiha, 2021). Therefore, such analysis might be misleading by not presenting the global picture. The proven dependency of VAWT performance on airfoil shape means that the design of morphing blades, which can adapt their shapes to variables such as azimuthal angle $\theta$ and tip speed ratio $\lambda$ is worth pursuing. In a smart rotor, as the blade profile morphs into a new geometry due to changes in azimuthal position or wind speed, the separation point will move to an optimal coordinate. As a result, flow detachment can be reduced or delayed to higher $\alpha$, and severe dynamic stall characteristics

can be controlled or even avoided in the case of unsteady separation at low $\lambda$, resulting in improved turbine performance (Tan and Paraschivoiu, 2017; Tirandaz and Rezaeiha, 2021).

Detailed analysis of the literature on morphing airfoils shows that the majority of studies focused on morphing trailing edges. For example, in an experimental study by (Pechlivanoglou et al., 2010), positive flap deflection was found to significantly increase lift force while negative flap deflection results in lift reduction, which is effective in rotor deceleration. A numerical study by (Wolff et al., 2014) has shown that morphing trailing edges, specifically the deflection angles and increasing length of the morphing trailing edge, have significant impact on lift force and thus, the stall characteristics of the blade. In another work by (Minetto and Paraschivoiu, 2020) a deformable trailing edge was discovered to alleviate the dynamic stall characteristics and improve the power output of VAWTs. (Tan and Paraschivoiu, 2017) showed that morphing the blade aileron to have the optimal shape for upwind and downwind quartiles can improve the aerodynamic performance of VAWTs. In addition, in a numerical study, it was found that changing the airfoil shape-defining parameters have a substantial impact on the power performance of VAWT operating in the dynamic stall regime (Tirandaz and Rezaeiha, 2021).

Despite the existence of this reported literature, several shape-defining parameters have received much less attention. Such parameters are hypothesised to have an influence on boundary layer events and the resultant aerodynamic loads. Therefore, a parametric analysis of these variables, with their potential to morph, would provide fundamental knowledge towards designing morphing blades for smart VAWTs.

**1.2 Objectives**

The present work follows the objectives below:

 i. To pave the road towards smart blades for VAWTs, having the capability of adaptation to different operational conditions.

 ii. To provide a set of generalizable conclusions from 630 transient simulations for 126 unique airfoils, generated with different values of maximum thickness $t/c$, chordwise position of maximum thickness $xt/c$, and leading-edge radius index $I$ at 5 different values of $\lambda$; and thus, understand the impact of different morphed-airfoil scenarios on the turbine power performance $C_P$ as well as the thrust performance $C_T$.

 iii. To prove the usefulness of the morphing technique as a promising tool to improve the power performance of VAWTs.

The reference airfoil is chosen from the symmetric modified NACA four-digit series. The modified airfoils are generated through changing the combination of the three aforementioned parameters. An unsteady Reynolds-Averaged Navier-Stokes (URANS) approach, previously validated with experimental data, will be used for the analysis. The results will provide a set of optimal airfoils at each $\lambda$, as well as each azimuth angle, and thus, making a conceptual step towards designing morphing blades for VAWTs.

**1.3 Paper outline**

The paper is organized as follows: Sect. 2 presents the computational settings and parameters for the simulations. The solution verification and validation studies are also included. Sect. 3 introduces the generated airfoil shapes. In Sect. 4, the results are presented in two scenarios. Sect. 5, 6, and 7 are devoted to the discussion, research limitations, and conclusions, respectively.

**2. Computational settings and parameters**

**2.1 Reference turbine characteristics**

A single-bladed Darrieus H-type VAWT was chosen as the reference case for this study (see Fig. 1 and Table 1). The turbine is a
simplified representation of the original one used by (Tescione et al., 2014). That is, the turbine shaft and spokes are removed, and
there is only one blade. Note that the conclusions are not significantly affected by these components. The reader is referred to our
earlier works (Rezaeiha et al., 2017b; Rezaeiha et al., 2018a) where it is shown that for low solidity VAWTs, the power
performance is almost independent of the shaft and number of blades. Therefore, such a simplified turbine model can effectively
reduce the computational costs of the huge number of simulations (i.e., 630 transient simulations) for the present work and, at the
same time, provide reliable results. Refs. (Rezaeiha et al., 2018a, b) are used to select the rest of the geometrical and operational
characteristics of the reference turbine.

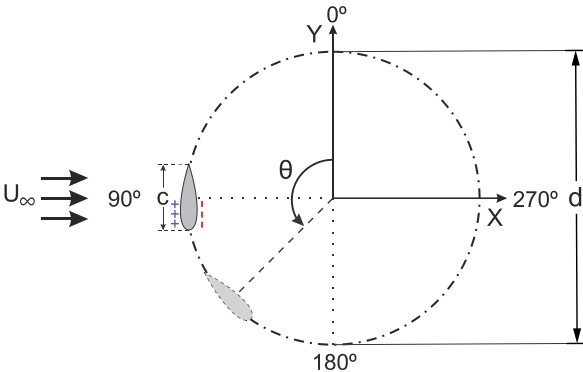

**Figure 1: The reference turbine (not to scale). (+): airfoil pressure side and (-): suction side for $0° \leq \theta < 180°$.**
**Table 1: Characteristics of the reference turbine.**

| Turbine type | Darrieus H-type |
|---|---|
| $n$ | 1 |
| $d$ | 1 m |
| $\sigma$ | 0.06 |
| Airfoil shape | NACA0018-6.0/3.0 (i.e., baseline NACA0018) |
| | $t/c = 18\%$; $I = 6.0$; $xt/c = 30\%$ |
| Blade/Spoke connection point | c/2 |
| $U_\infty$ | 9.3 m/s |
| $\lambda$ | 2.5, 3.0, 3.5, 4.5, 5.0 |
| $\Omega$ | 46.5, 55.8, 65.1, 83.7, 93.0 rad/s |
| $c$ | 0.06 m |
| $Re_c$ [$\times 10^5$] | 1.03, 1.20, 1,40, 1.76, 1.95 |
| $TI$ | 5% |

**2.2 Computational settings**
The commercial flow solver ANSYS Fluent v2019R2 is employed for the 2D incompressible URANS simulations coupled with
the four-equation transition SST turbulence model. The simulations are solved using second-order spatial/temporal discretization
and the SIMPLE pressure-velocity coupling scheme. The computational domain, grid, and boundary conditions are summarized
in Table 2. The schematic of the computational domain and the computational grid and its subregions are shown in Fig.2.
Some attempts have been made to identify the proper computational settings for the simulation of H-type Darrieus turbine
(Balduzzi et al., 2016a; Balduzzi et al., 2016b). However, in this work, the turbulence model is selected based on our previous
findings (Rezaeiha et al., 2019b, 2020a). Best-practice guidelines for the CFD simulations of VAWTs are used to select the domain
size, the azimuthal increment, and the convergence criterion (Rezaeiha et al., 2018c). The corresponding absolute time-step values
are $3.75339546\times10^{-5}$ s, $3.12782955\times10^{-5}$ s, $2.68099676\times10^{-5}$ s, $2.0852197\times10^{-5}$ s and $1.70608885\times10^{-5}$ s for $\lambda = 2.5, 3.0, 3.5, 4.5$
and 5.5, respectively. With the selected $d\theta = 0.1$, 3600 time-steps per turbine revolution are achieved. A total number of 20
revolutions, i.e., 72,000 time-steps, are simulated before the results of the present study are obtained at the 21st turbine revolution.
Under these conditions, the statistical convergence of the transient simulations is ensured. In each case, a number of 20 iterations
per time-step is performed so that the scaled residuals stay $< 10^{-5}$.

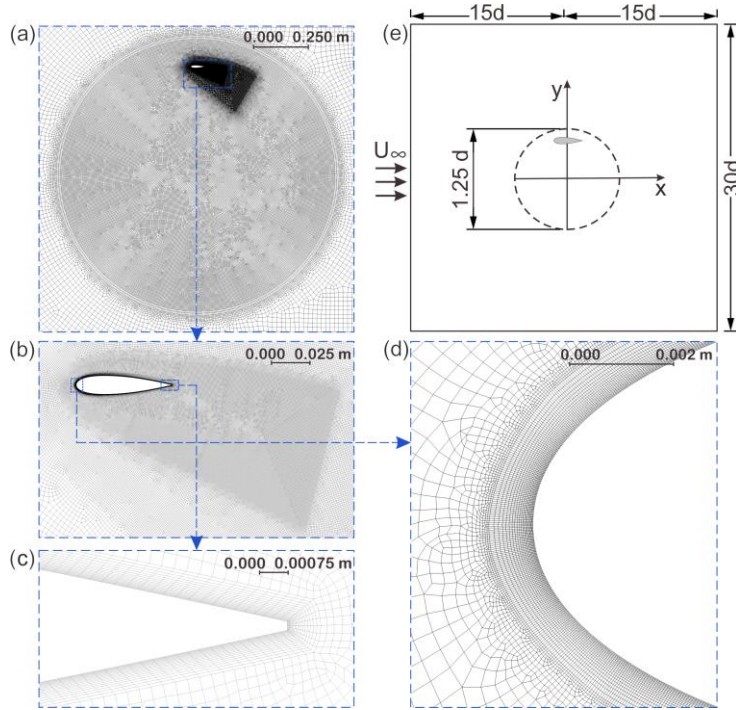


**Figure 2. (a-d) the grid; and (e) schematic of the computational domain (not to scale).**
**Table 2: Details of computational domain, grid, and boundary conditions.**

| | |
|---|---|
| Computational domain (see Fig. 2e) | $30d \times 30d$<br>(*d*: turbine diameter) |
| Computational grid (see Fig. 2a-d) | *Cell type*: quadrilateral<br>*Cell No.*: 302,815<br>*No. of cells around the airfoil circumference*: 800<br>$y^+_{max} < 2.5$ |
| Boundary conditions | *Inlet*: uniform normal velocity (Turbulence length scale = *d*);<br>*Outlet*: zero static gauge pressure; |

**2.3 Solution verification and validation**
The domain type is selected based on our earlier studies, where the difference between 2D and 2.5D URANS simulations was
found to be insignificant (Rezaeiha et al., 2017a). A grid convergence analysis using uniformly-doubled grids has been performed
and documented in Ref. (Rezaeiha et al., 2019c), which for brevity is not repeated here. Three experimental studies with different
test conditions previously were used to validate the CFD simulations. The different geometrical and operational characteristics of
the turbines used in the experiments led to dissimilar conclusions (Tescione et al., 2014; Ferreira et al., 2009; Castelli et al., 2011),
ensuring a high level of confidence in the accuracy of the CFD simulations. However, the reader is referred to Ref. (Rezaeiha et
al., 2019b) for more detailed descriptions of the validation studies.
**3. Airfoil shape modification**
Figure 3 shows a schematic drawing of the symmetric modified NACA 4-digit airfoil and the selected shape-defining parameters
for this study. These parameters are modified within their most common regimes as follows:
(i)   relative maximum thickness (*t/c*): 10, 12, 15, 18, 21 and 24%;

(ii)  relative chordwise position of maximum thickness (*xt/c*): 20, 22.4, 25, 27.5, 30, 35 and 40%;

(iii) index of leading-edge radius (*I*): 4.5, 6.0 and 7.5.

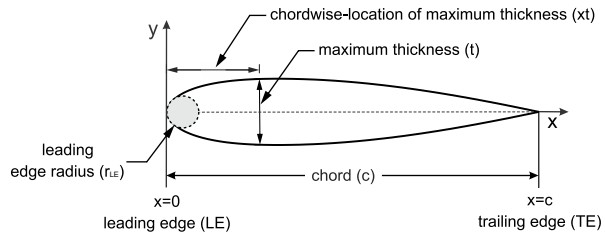


**Figure 3: Defining parameters of the symmetric airfoil.**


**Figure 4: Studied airfoil shapes.**

Note that any value of *I* out of the selected range results in a too sharp or too blunt leading edge. The analysis is based on 126
airfoil shapes (see Fig. 4). The modification of the airfoil coordinates and the related equations are documented in Ref. (Tirandaz
and Rezaeiha, 2021). The focus of this study is on symmetric airfoils with zero camber. The modified airfoils are designated as the
*NACA00t/c – I / xt/c*. The first symbol from left to right, i.e., *t/c*, represents the maximum thickness in %c; the second one, *I*, shows
the index of leading-edge radius (with one decimal precision); and the third one, *xt/c,* is the chordwise position of the maximum
thickness in $10^{th}$ of the chord with two decimal precision. For example, the NACA0024-4.5/3.50 has a maximum thickness of *t/c*
= 24%, located at *xt/c* = 35%, and a leading-edge radius index of *I* = 4.5.
**4. Results**
The results are presented in two scenarios, namely, optimal airfoils as functions of $\lambda$ (Sect. 4.1), and d$\theta$ (Sect. 4.2). In Sect. 4.3 the
performance of the optimal airfoils from the first scenario are compared with that of the reference airfoil. A coupled analysis is
performed at different $\lambda$ of 2.5, 3.0, 3.5, 4.5 and 5.5. Figure 5 depicts the variations of $\alpha$ as the turbine passes through its last
revolution. Note that the higher the value of $\lambda$ is, the more limited the variations of $\alpha$ are. For $\lambda$ = 2.5, 3.0 and 3.5, the variations of
$\alpha$ exceed the $\alpha_{ss}$ for all the studied airfoils; while at higher $\lambda$ = 4.5 and 5.5, this behaviour is not observed for all of the studied
airfoils. The reader is referred to (Rezaeiha et al., 2018b), where the method of calculating the $\alpha$ from the CFD results is provided
in detail. However, in a recent study by (Melani et al., 2020) an ad hoc inverse verification procedure was developed to compare
the accuracy of three selected methods in calculating the angle of attack from the CFD flow field, including 3-Points, Line Average,
and Trajectory approaches.

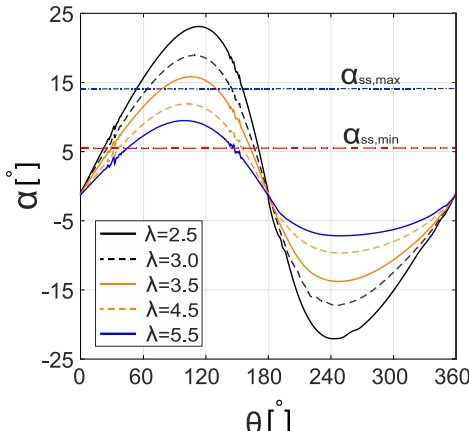

**Figure 5: $\alpha$ versus $\theta$ for different $\lambda$. The $\alpha_{ss,min}$ and $\alpha_{ss,max}$ are based on Xfoil.**
**4.1  Modification of the airfoil shape-defining parameters**
To derive the optimal airfoil for each $\lambda$, the combination of the $t_{opt}/c$, $xt_{opt}/c$, and $I_{opt}$, corresponding to the turbine $C_{P,max}$ is
determined. Sect. 4.1.1 to 4.1.3 are devoted to the discussions on individual modification, and Sect. 4.1.4, presents an overall view
on the combined modification of the aforementioned parameters.
**4.1.1 Modification of the maximum thickness ($t/c$)**
Figure 6 shows the impact of changing $t/c$ on the turbine $C_P$ for the studied range of $xt/c$, $I$ and $\lambda$. Figure 7 shows the instantaneous
moment coefficient $C_m$ versus $\theta$ for selected $t/c$ and $xt/c$. It can be observed that:
*Regarding the lowest value of I = 4.5* (see Fig. 6a – e and Fig. 7): Generally speaking, the trend of $C_P$ – $t/c$ for different $\lambda$ is similar,
except for some noticeable differences. That is, by increasing $\lambda$, the $C_P$ shows higher sensitivity to $t/c$. This is reflected as higher
$|\Delta C_P|$ and can be explained by the following: by changing $t/c$, the pressure gradient changes over the airfoil; therefore, the transition
point, the separation and stall characteristics, and eventually the resultant aerodynamic loads also change. However, when the flow
is fully separated in the post-stall regime, changing $t/c$ has no longer significant impact on $C_P$. By increasing $\lambda$, and thus, more
limited variation of $\alpha$, the blade passes over a range of fewer azimuth angles in the post-stall regime (see Fig. 5). Due to this,
changing the $t/c$ is influential within a wider range of effective $\theta$. This can be recognized by the improved $C_P$ for higher $\lambda$. At $\lambda$ =
2.5, the $C_P$ follows a non-monotonic trend for $xt/c \leq 30\%$, and a monotonic upward trend for $xt/c \geq 35\%$. Nevertheless, with the
exception of $xt/c \leq 22.5\%$ at $\lambda$ = 5.5, where the $C_P$ monotonically decreases by increasing $t/c$, the trend remains non-monotonic for
different values of $xt/c$ at the studied range of $\lambda$. That is, by changing the $t/c$ to higher values, the $C_P$ experiences an initial growth
to its maximum value at $t_{opt}/c$, followed by a reduction for $t/c > t_{opt}/c$. This can be recognized from the $C_m$ plots, where by changing
$t/c$ to its optimal value at $t_{opt}/c$, the sudden drop in $C_{m,max}$, which indicates the instant of moment stall, is observed at higher $\theta$; the
consequent fluctuation is alleviated, and the mean value of $C_m$ increases, thus, making consistency with the highest $C_P$ at $t_{opt}/c$ for

a fixed *xt/c* ( see Fig. 7a-i for selected *xt/c*). This can be attributed to the following observations from the skin friction, lift, and drag coefficients ($C_f$, $C_l$, and $C_d$): when the turbine is operating at low values of $\lambda \leq 3.5$, increasing *t/c* from 10% to $t_{opt}/c$, changes the stall type from mixed stall for *t/c* = 10% to trailing-edge stall for thicker airfoils; an earlier formation of laminar separation bubble (LSB) and trailing-edge separation (TES) is observed; TES-LSB merging (i.e., full-flow separation) is discovered to occur at higher azimuth, indicating a more extended favorable pressure gradient for $t_{opt}/c$ (see for example Fig. 8 for *xt/c* = 27.5% at $\lambda$ = 2.5); lighter dynamic stall is observed; that is, lift and drag jump, which indicate the onset of dynamic stall, reduce and shift to higher azimuth, and the consequent post-stall loads fluctuations are alleviated (see for example Fig. 9 for *xt/c* = 27.5% at $\lambda$ = 2.5). However, an earlier stall is found to occur for *t/c* > $t_{opt}/c$ due to more pronounced earlier merging of TES-LSB. This is reflected by lower $C_P$ and $C_{m,max}$ for *t/c* > $t_{opt}/c$ (see Figs. 6a-c and 7a-i). Note that the monotonic growth in $C_P$ – *t/c* for *xt/c* $\geq$ 35% at $\lambda$ = 2.5 can also be explained with the aforementioned reasoning, yielding the $C_{P,max}$ at the highest thickness of $t_{opt}/c$ = 24% (see Fig. 6a).

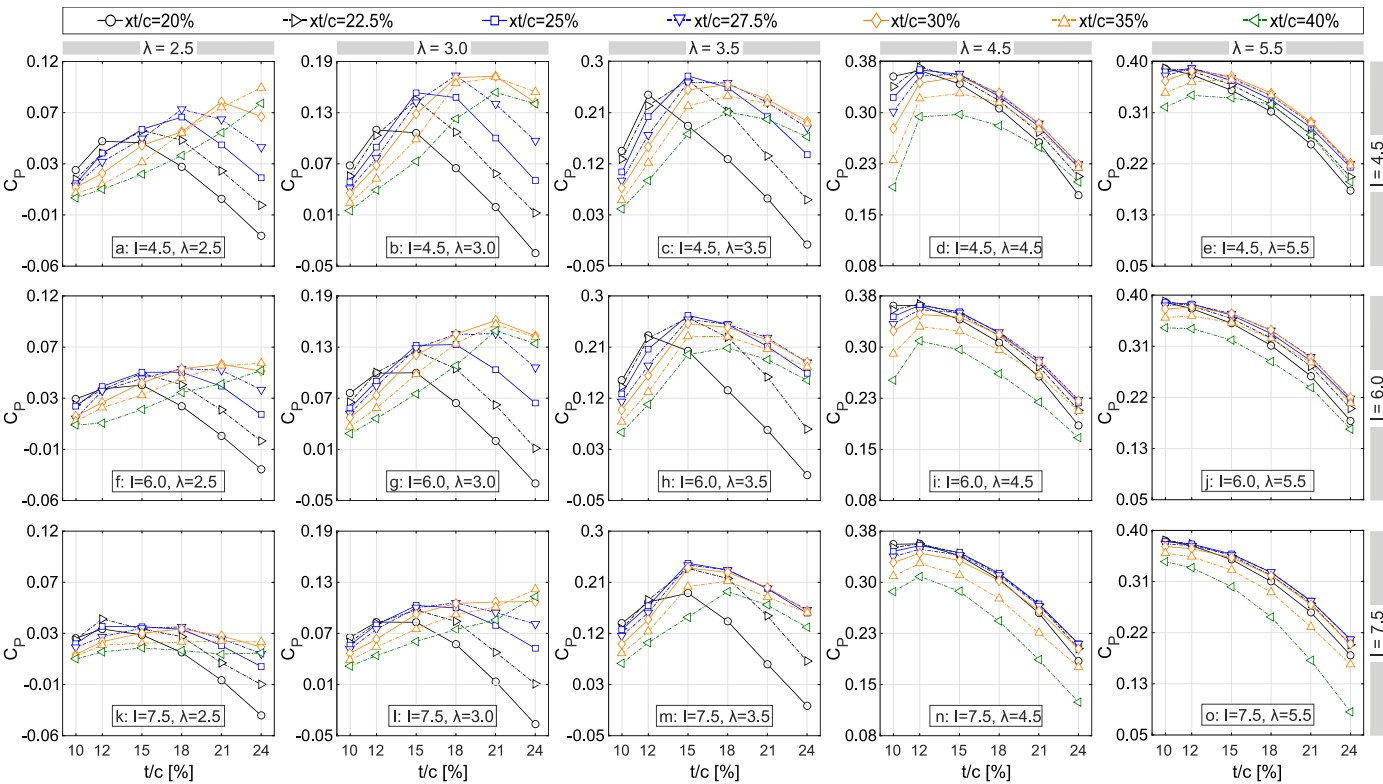

**Figure 6: Impact of changing *t/c* on the turbine $C_P$ at different *xt/c* and $\lambda$.**

Table 3 shows the $t_{opt}/c$, corresponding to each *xt/c* (i.e., $t_{opt,xt}/c$) at different $\lambda$. The $t_{opt}/c$ corresponding to each $\lambda$ is indicated by a star sign. It can be seen that by increasing *xt/c*, which means a longer favorable pressure gradient on the blade, a higher thickness is needed for the airfoil to be optimal. Note that, increasing $\lambda$, influences the shape of the optimal airfoil by decreasing its thickness. In other words, the higher $\lambda$ is, the thinner the optimal airfoil is. This is consistent with the findings documented in (Healy, 1978; Subramanian et al., 2017), where it shows the superior performance of thick airfoils at low $\lambda$. This may be attributed to the turbine operational regime as follows: When the turbine goes into regimes with higher $\lambda$ and more pronounced reduction in the variation of $\alpha$, higher values of $C_l$ at lower $\alpha$ is of most impact on the turbine $C_P$. Therefore, thinner airfoils with a higher lift curve slope outperform the thicker ones with a lower slope of the $C_l$ – $\alpha$. Eventually, this results in less pronounced sensitivity of the $t_{opt}/c$ to *xt/c*, and shifting the peak in $C_P$ – *t/c* (i.e., $t_{opt}/c$) towards the lowest *t/c* = 10% and 12% in the non-dynamic stall regime with $\lambda \geq$ 4.5 (see Table 3). The analysis also shows a drag increment for thicker airfoils at $\lambda$ = 4.5 and 5.5, which is a result of the earlier formation of LSB and TES. This is consistent with the reduction in $C_P$ and $C_{m,max}$ for *t/c* > $t_{opt}/c$ (see Figs. 6d-e and 7j-o). Note that

for $xt/c \leq 22.5\%$ at $\lambda = 5.5$ the same reasoning results in a monotonic decrease of $C_P$, yielding the $C_{P,max}$ at the lowest thickness of
$t/c = 10\%$.
The effect of flow curvature on aerodynamic loading is another important physical phenomenon to take into account in predicting
the performance of VAWTs. Because of the angular velocity of the turbine rotor blades, the relative flow direction continuously
varies along the airfoil chord, and thus, the blades experience curved streamlines. As a result of this, a symmetrical airfoil with
zero pitch angle in the circular path of a VAWT rotor behaves as if it's a cambered airfoil with a non-zero pitch angle in a straight
flow (Migliore et al., 1980; Rainbird et al., 2015). The flow curvature effects become less pronounced on a curved airfoil (Coiro
et al., 2005). In addition, a blade hinge located at 50% chord length significantly alleviates the flow curvature effects. However,
among all the parameters, the ratio of blade chord to turbine rotor radius ($c/R$) has the greatest impact on flow curvature effects
(Migliore et al., 1980). For low values of $c/R$ (i.e., low solidity), the blade surface pressure distribution shows negligible differences
with respect to that of the no-lift condition (Coiro et al., 2005), indicating less pronounced effects of flow curvature on the
performance of low-solidity turbines (Rainbird et al., 2015). In this study, due to the low value of $c/R = 0.12$ (i.e., low $\sigma$), the
contribution of flow curvature effects is considered to be small.
*Regarding the moderate and highest values of $I = 6.0$ and $7.5$* (see Fig. 6f-j and 6k-o): The overall trend for $C_P$ is very similar to
that of the lowest $I = 4.5$; however, it shows comparatively lower values of $|\Delta C_P|$), especially for $\lambda \leq 3.5$. The impact of changing
the $r_{LE}$ on the turbine $C_P$ is separately discussed in detail in Sect. 4.1.3; therefore, it is not included here.
**Table 3: $t_{opt,xt}/c$ for different values of $xt/c$ and $\lambda$ ($I = 4.5$).**

| $\lambda$ | 20 | 22.5 | 25 | 27.5 | 30 | 35 | 40 | $xt/c$ [%] |
|---|---|---|---|---|---|---|---|---|
| 2.5 | 12 | 15 | 18 | 18 | 21 | 24* | 24 | |
| 3.0 | 12 | 15 | 15 | 18* | 21 | 21 | 21 | |
| 3.5 | 12 | 15 | 15* | 15 | 18 | 18 | 18 | $t_{opt,xt}/c$ [%] |
| 4.5 | 12 | 12* | 12 | 15 | 15 | 15 | 15 | |
| 5.5 | 10 | 10* | 12 | 12 | 12 | 15 | 12 | |
| * $t_{opt}/c$ at the corresponding $\lambda$ | | | | | | | | |

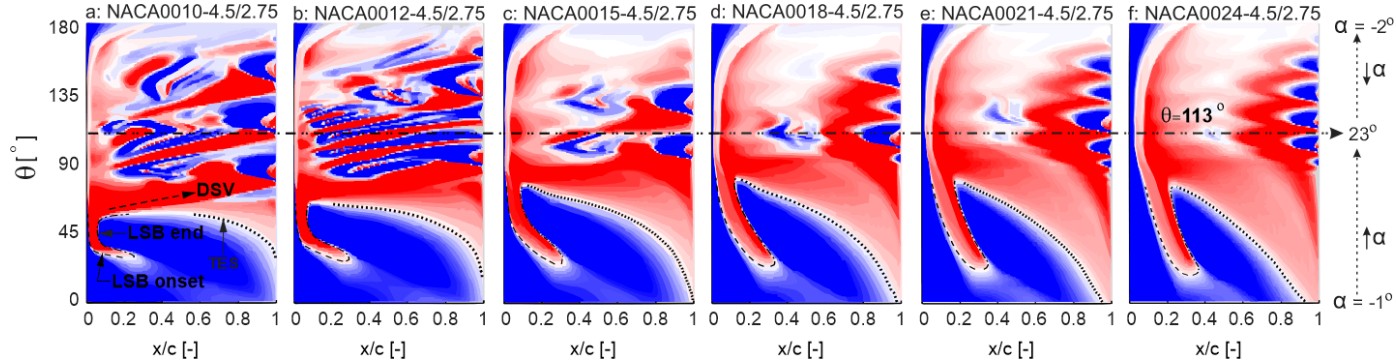

**Figure 7: Impact of changing *t/c* on the turbine $C_m$ for selected *xt/c* and *t/c* at different $\lambda$.**

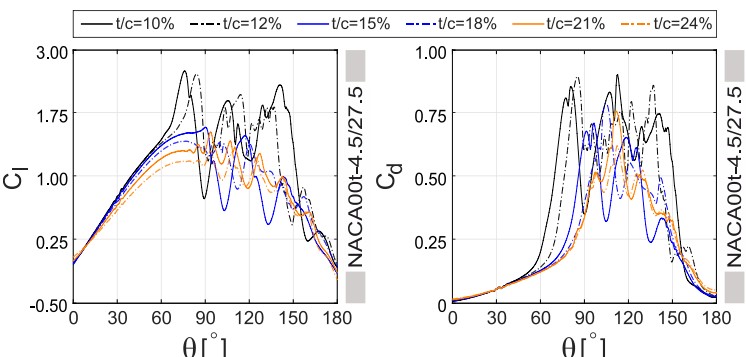

**Figure 8: Spatiotemporal contour plots of $C_f$ along the suction side of the blade during the first-half of the last revolution for the NACA00*t*-4.5/27.5 at $\lambda$ = 2.5. Note that the X-axis is along the chord line and $\theta$ = 113° corresponds to the blade's $\alpha_{max}$ = 23°.**

**Figure 9: Impact of *t/c* on variations of *$C_l$* and *$C_d$* versus *θ* during the first-half of the turbine last revolution for *xt/c* = 27.5% and *I* = 4.5 at *λ* = 2.5.**

### 4.1.2 Modification of the chordwise position of maximum thickness (*xt/c*)

Figure 10 shows the variation of the turbine *$C_P$* versus *xt/c* at the studied range of *t/c*, *I*, and *λ*. Figure 11 shows the instantaneous moment coefficient *$C_m$* versus azimuth for selected *xt/c* and *t/c*. It can be seen that:

*Regarding the lowest value of I = 4.5 (ee Fig. 10a-e):* The overall trend of *$C_P$ – xt/c* for different *λ* is very similar, except for the following differences. By increasing *λ*, the turbine *$C_P$* shows higher |Δ*$C_P$*|. This is due to the similar reasoning discussed earlier in Sect. 4.1.1, and summarized as follows: changing the *xt/c* results in changing the boundary layer and stall characteristics. On the other hand, increasing *λ* is associated with lower variation of *α*, i.e., a more limited azimuthal range of the post-stall regime. As a result, the impact of changing the *xt/c* becomes significant over a wider range of *θ*, resulting in improved *$C_P$*.

For *t/c* ≤ 12% in the dynamic stall regime with *λ* ≤ 3.5, the *$C_P$* monotonically decreases by increasing the *xt/c*, yielding the *$C_{P,max}$* with the lowest *xt/c* of 20% (see Fig. 10a-c). However, apart from *t/c* = 10% at *λ* = 4.5, where *$C_P$* monotonically decreases, the trend for thin airfoils changes to non-monotonic at *λ* ≥ 4.5 (see Fig. 10d-e). In other words, by increasing the *xt/c* from 20% to 40%, the *$C_P$* grows to its maximum value at *$xt_{opt}/c$*, before decreasing for *xt/c* > *$xt_{opt}/c$*. The monotonic behavior of *$C_P$* for thin airfoils at low *λ* can be explained based on the observations of the skin-friction coefficient *$C_f$* as follows: The dynamic stall for *t/c* ≤ 12% is preceded by either (i) gradual extension of the LSB towards the trailing edge (thin-airfoil stall), or (ii) a sudden upstream propagation of the TES (leading-edge stall). Changing the *xt/c* to higher values results in either an earlier downstream extension of the LSB, or an earlier formation and abrupt-upstream propagation of the TES; and consequently, an advanced stall on the blade. This is evident from the *$C_m$* plots for *t/c* = 12% (see Fig. 11a-c), where the abrupt drop in *$C_{m,max}$* occurs at a lower *θ*, indicating an earlier moment stall due to increasing the *xt/c*. The overall lower values of *$C_m$* for higher *xt/c* justify the monotonic reduction in *$C_P$*. For brevity, the *$C_f$* plots are not presented here.

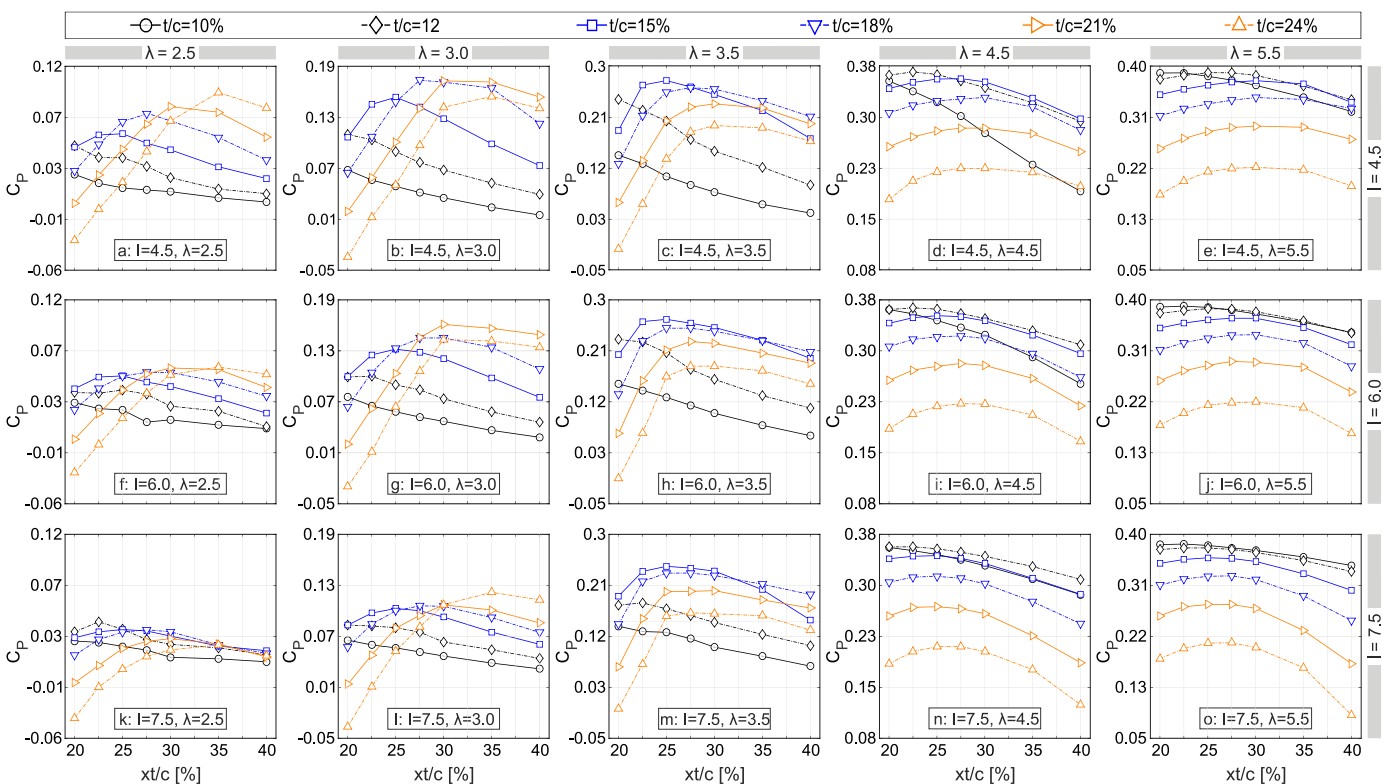

**Figure 10: Impact of changing *xt/c* on the turbine *C_P* at different *t/c* and *λ*.**

On the other hand, the non-monotonic trend of $C_P$ for thin airfoils at $λ ≥ 4.5$ (i.e., non-dynamic stall regime) can be recognized
from the $C_m$ plots. For example, by changing the $xt/c$ from 20% to $xt_{opt}/c = 25\%$ for $t/c = 12\%$ $λ = 5.5$, the $C_{m,max}$ slightly increases
before decreasing for $xt/c ≥ 27.5\%$ (see Fig. 10e). This can be explained by the skin-friction coefficient $C_f$, where it shows an
earlier formation and upstream propagation of the TES, and thus, a promoted TES for $xt/c > xt_{opt}/c$ (see Fig. 12). Note that, when
the adverse effects of dynamic stall are suppressed at $λ ≥ 4.5$, increasing $xt/c$ shows a marginal positive impact on the $C_P$ for thin
airfoils, reflecting a non-monotonic trend of $C_P$ versus $xt/c$. However, the value of $t/c$ for thin airfoils plays a more crucial role in
this regime. This can be observed from the sharp downward trend of $C_P – xt/c$ for $t/c = 10\%$ at $λ = 4.5$, while it changes to a non-
monotonic trend for $t/c = 12\%$. This may be attributed to the more pronounced formation and propagation of TES, and thus, an
earlier stall due to increasing $xt/c$ for $t/c = 10\%$. However, the trend of $C_P – xt/c$ for $t/c$ of 10% remains non-monotonic at $λ = 5.5$,
showing less sensitivity to TES at higher $λ$.

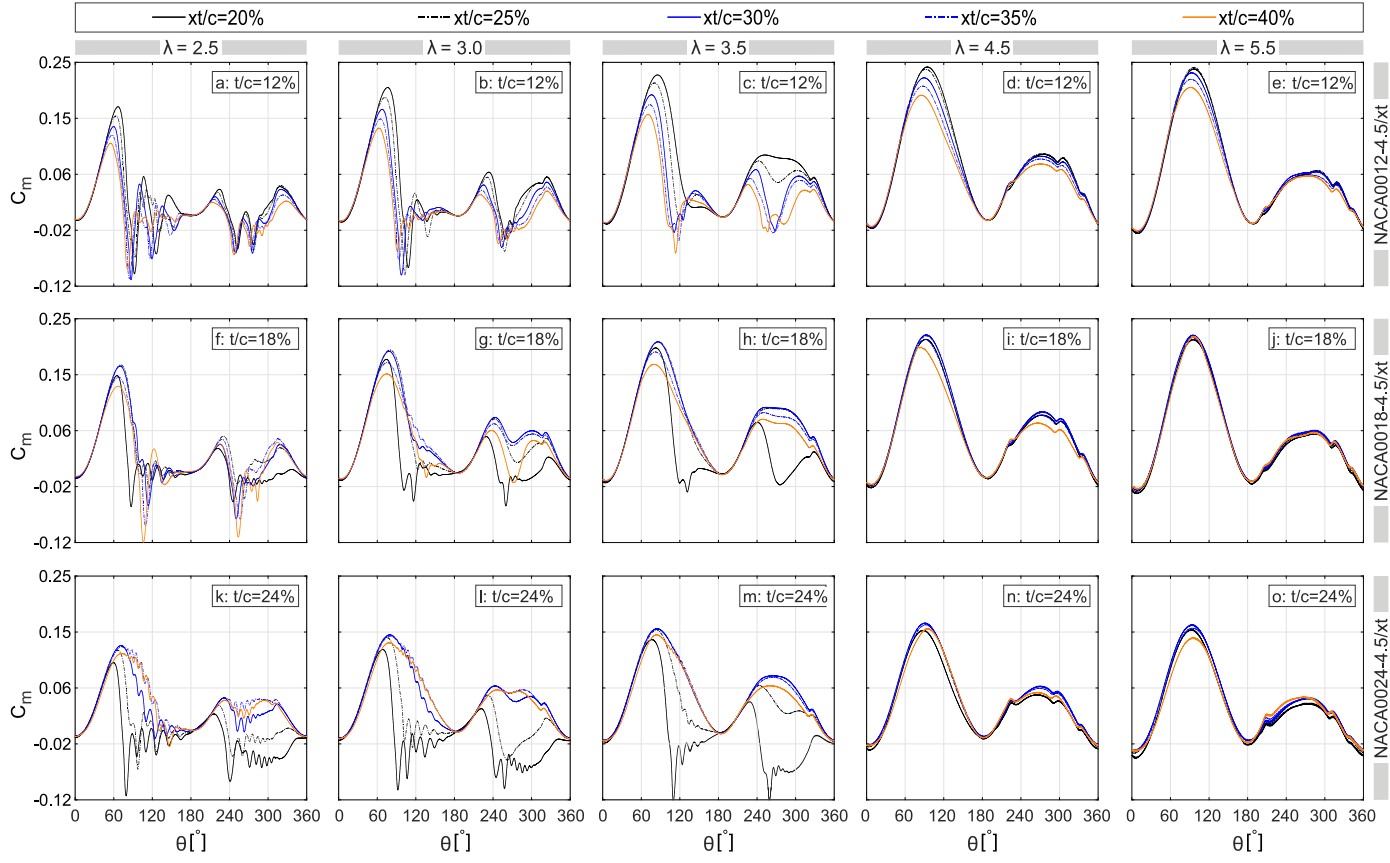


**Figure 11: Impact of changing *xt/c* on the turbine *C_m* for selected *t/c* and *xt/c* at different *λ*.**

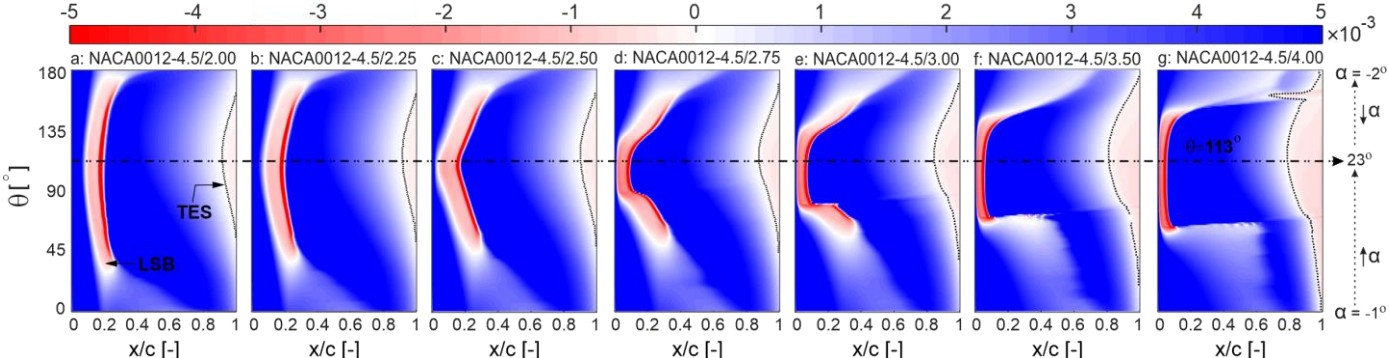


**Figure 12: Spatiotemporal contour plots of $C_f$ along the suction side of the turbine blade during the first-half of the last revolution for the NACA0012-4.5/$xt$ at $\lambda = 5.5$. Note that the X-axis is along the chord line and $\theta = 113°$ corresponds to the blade's $\alpha_{max} = 23°$.**

For the medium- and high-thickness airfoils (i.e., $t/c \geq 15\%$), the turbine $C_P$ follows a trend with a defined maxima at $xt_{opt}/c$ (see Figs. 10a-e). As previously discussed in Sect. 4.1.1, this non-monotonic trend is a consequence of thicker-airfoil stall type, which is triggered by the formation of a flow reversal near the trailing edge (Mccroskey, 1981; Sharma and Visbal, 2019; Frolov, 2016; Meseguer et al., 2007). Therefore, when $xt/c$ changes to its optimal value, the adverse pressure gradient becomes less severe, resulting in improved stall characteristics. This can be recognized by either dynamic stall alleviation at low values of $\lambda \leq 3.5$, or a postponed stall at non-dynamic stall regimes with $\lambda \geq 4.5$. Table 4 gives the $xt_{opt,t}/c$ (i.e., the $xt_{opt}/c$ at each $t/c$) in terms of $C_{P,max}$ for each $\lambda$. The corresponding $xt_{opt}/c$ for different $\lambda$ is indicated by a star sign. For $\lambda \leq 3.5$, by increasing $t/c$, the $xt_{opt,t}/c$ also increases. However, by increasing $\lambda$ from 2.5 to 3.5, and thus, encountering a comparatively lighter dynamic stall and more limited variation of $\alpha$, the $xt_{opt}/c$ and its corresponding $t/c$ decrease (see also Fig. 10). The reason for the outperformance of thin airfoils at higher $\lambda$ is explained earlier in Sect. 4.1.1. Nevertheless, in the dynamic stall regime, the outperformance of moderate to high values of $xt/c$ for thicker airfoils at a fixed $\lambda$ is readily apparent from the turbine $C_m$ for selected $t/c = 18\%$ and $24\%$ (see Fig. 11f-h and k-m). It can be seen that increasing $xt/c$ to its optimal value results in an increase in the $C_m$ curve peak, a delay in the sudden drop of $C_{m,max}$, less pronounced subsequent fluctuations, and higher values of $C_m$ in the turbine downwind quartile. This is due to alleviated dynamic stall, and is more pronounced for $t/c = 24\%$ (see Fig. 11k-m). A further increase in $xt/c > xt_{opt}/c$, is found to have a negative effect on $C_m$ and finally leads to an earlier stall. This is because increasing the $xt/c$ higher than $xt_{opt}/c$ promotes the formation of LSB and TES, and results in an earlier full-flow separation and drop in $C_{l,max}$. Please note that for better illustration, the $C_m$ plots are not presented for all the studied values of $xt/c$. For $\lambda \geq 4.5$, by increasing $xt/c$ for $t/c \geq 15\%$, the $C_P$ shows less sensitivity to $xt/c$ and the corresponding $xt_{opt}/c$ changes marginally (see Fig. 10d-e and Table 4). This is consistent with the turbine $C_m$ plots for selected $t/c = 18\%$ and $24\%$, where the $C_{m,max}$ and the azimuth of moment stall are almost invariant to $xt/c$ (see Fig. 11i-j and Fig. 11n-o).

*Regarding the moderate and highest values of I = 6.0 and 7.5 (see Fig. 10f-j and 10k-o):* The $C_P – xt/c$ shows a similar trend to that of $I = 4.5$. However, in dynamic stall regime (i.e., $\lambda \leq 3.5$), the turbine $C_P$ shows a considerably smaller $|\Delta C_P|$, especially for higher $xt/c$. On the other hand, in non-dynamic stall regime with $\lambda \geq 4.5$, a marginal reduction in $|\Delta C_P|$ is observed. However, the $C_P – xt/c$ shows more pronounced sensitivity to changing $xt/c$ for the moderate and thick airfoils.

**Table 4: $xt_{opt,t}/c$ for $I = 4.5$ at different $t/c$ and $\lambda$.**

| $\lambda$ | 10 | 12 | 15 | 18 | 21 | 24 | $t/c$ [%] |
|---|---|---|---|---|---|---|---|
| 2.5 | 20 | 20 | 25 | 27.5 | 30 | 35* | |
| 3.0 | 20 | 20 | 25 | 27.5* | 30 | 35 | |
| 3.5 | 20 | 20 | 25* | 27.5 | 30 | 30 | $xt_{opt,t}/c$ [%] |
| 4.5 | 20 | 22.5* | 27.5 | 30 | 27.5 | 27.5 | |
| 5.5 | 22.5* | 25 | 30 | 30 | 30 | 30 | |
| * $xt_{opt}/c$ at the corresponding $\lambda$ | | | | | | | |

### 4.1.3 Modification of the leading-edge radius ($r_{LE}$)

Figure 13 shows the impact of changing $r_{LE}$ on the $C_P$ for selected airfoils at different $\lambda$. Figure 14 shows a comparison of the $C_P – xt/c$ for different $I$ and selected values of $t/c$. The analysis is grouped based on the maximum thickness as follows:

*Regarding the thin airfoils (t/c = 10% and 12%) (see Fig. 13 and 14a-e):* regardless of $xt/c$, the turbine $C_P$ is marginally influenced by the $r_{LE}$. This can be attributed to the low dependency of thin airfoils and the relevant aerodynamic loads on $r_{LE}$, which is due to the geometrical constraints imposed by the airfoil thickness. It can be observed that by increasing the index of $r_{LE}$ for different $xt/c$

at $\lambda \leq 3.5$, $C_P$ slightly changes; this minimal difference is in line with the corresponding $C_m$ plots for $t/c \leq 12\%$. This can also be
recognized from the skin friction, lift, and drag coefficients by the negligible changes in the characteristics of boundary layer
events, including LSB and TES, and consequently the onset of dynamic stall and $C_{d,max}$. Due to the large volume of the results, the
$C_m$, $C_l$, $C_d$, and $C_f$ plots are not presented here. For $\lambda \geq 4.5$, except for the NACA0010-$I$/3.5, where increasing $r_{LE}$ has the most
influence on $C_P$, the aerodynamic loads and the turbine $C_m$ show even less sensitivity to $r_{LE}$. Note that this is the regime in which
the dynamic stall is no longer encountered and thin airfoils outperform the rest of the airfoils. The impact of $r_{LE}$ on the turbine $C_P$
for the optimal thin airfoils at $\lambda \geq 4.5$ is shown in Fig. 13f. Figures 15d and e show the corresponding $C_m$ plots.
*Regarding the moderately-thick airfoils (t/c = 15% and 18%)* (see Fig. 13 and 14f-j): overall, the turbine $C_P$ shows higher
dependency and sensitivity to $r_{LE}$. The higher dependency is due to the less severe geometrical constraints imposed by the
moderately thick airfoils. Thus, changing the $r_{LE}$ noticeably modifies the airfoil shape and thereby influences the aerodynamic
loads. The higher sensitivity is reflected by the noticeable monotonic reduction of $C_P$ for most of the $xt/c$ values. This significant
decrease can be recognized from the $C_m$ plots, where the curve peak drops by increasing the leading edge radius index. This may
be due to the promoted LSB and TES characteristics, which result in higher $C_{d,max}$ for larger $r_{LE}$. For $\lambda \leq 3.0$, the more prominent
sensitivity is observed within the range of $22.5\% \leq xt/c \leq 35\%$; however, the $C_P$ shows less sensitivity to $r_{LE}$ for $\lambda = 3.5$,
corresponding to a lighter dynamic stall regime (see Fig. 14f-h). Note that the moderately thick airfoils show superior performance
over the thin and thick airfoils at $\lambda = 3.0$ and 3.5 (i.e., the NACA0018-4.5/2.75 and NACA0015-4.5/2.5, respectively). Figure 13f
and Figs. 15b and c show the impact of changing $r_{LE}$ on the turbine $C_P$ and $C_m$ for the optimal airfoils at $\lambda = 3.0$ and 3.5. When the
turbine goes into the non-dynamic stall regime with $\lambda \geq 4.5$, the range of $xt/c$ within which the index of leading-edge radius is the
most influential, shifts downstream to $30\% \leq xt/c \leq 40\%$ (see Fig. 14i and j).

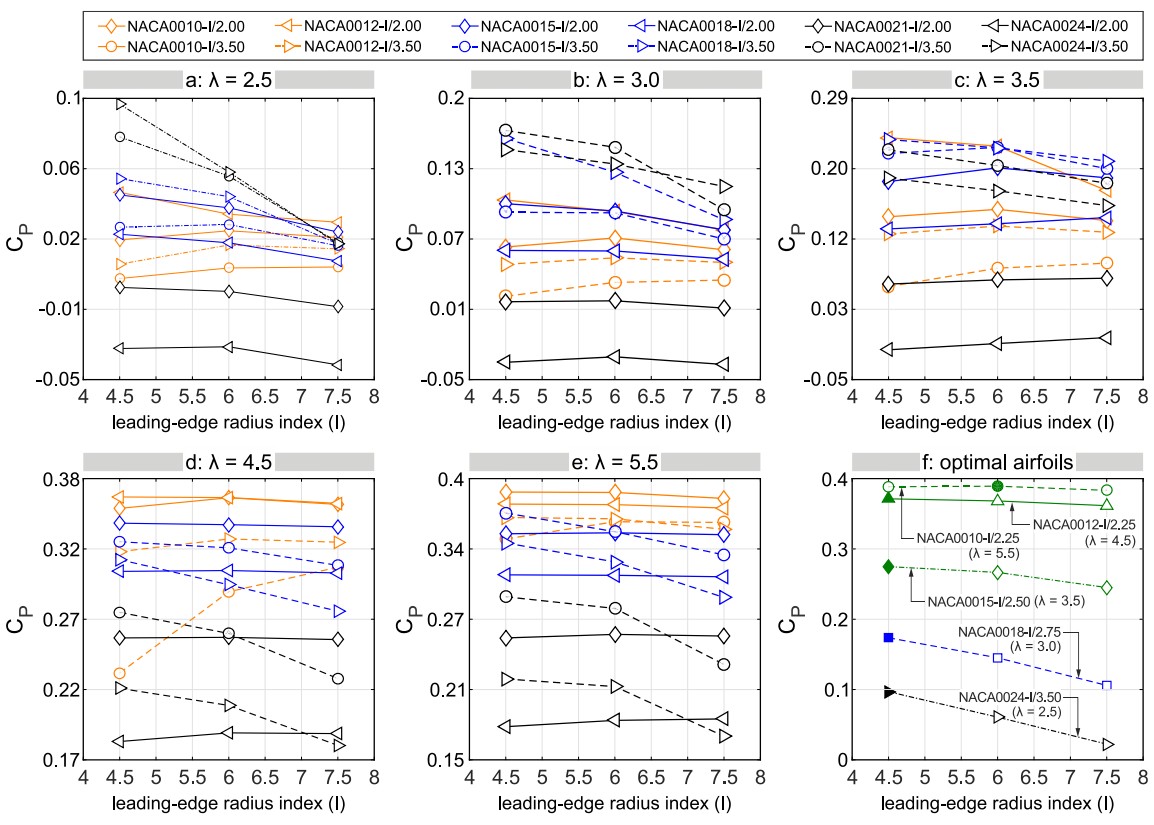

**Figure 13: Impact of changing $r_{LE}$ on the $C_P$ for (a-e) selected, and (f) optimal airfoils at different $\lambda$. Filled symbols represent the**
**optimal airfoils corresponding to each $\lambda$.**
*Regarding the thick airfoils (t/c = 21% and 24%)* (see Fig. 13 and 14k-o): the analysis shows that at $\lambda = 2.5$, thick airfoils
significantly surpass other airfoils in terms of power performance (see Fig. 13). Aside from the following differences, the overall
trend of $C_P - xt/c$ is quite similar to that of moderately thick airfoils: $C_P$ values are more sensitive to $r_{LE}$ at $\lambda = 2.5$, 4.5 and 5.5, but
less so at $\lambda = 3.0$ and 3.5 (see Fig. 14k-o). By increasing the $r_{LE}$, the $C_P$ values experience a monotonic reduction, especially for
thick airfoils with $xt/c \geq 30\%$ at $\lambda = 2.5$, where the variation of $I$ is the most influential on $C_P$. For example, the overall reduction
of $C_P$ for the NACA0024-$I$/3.50 at $\lambda = 2.5$, 3.0, 3.5, 4.5 and 5.5 is 77%, 21%, 17%, 19% and 23%, respectively. This can be
recognized from the $C_m$ plots, where the $C_m$ values decrease dramatically in both upwind and downwind quartiles, the $C_m$ curve
peak drops, and the post-stall $C_m$ fluctuation gets more significant (see Fig. 15a). This is due to earlier formations of the LSB and
TES, and thus a higher $C_{d,max}$. Thick airfoils with low $xt/c$ show marginal sensitivity to $r_{LE}$ at different $\lambda$. The corresponding $C_m$
plots show approximately the same azimuth of moment stall for different $I$. For brevity, the $C_m$ plots are only presented for the
NACA0024-$I$/3.50, which is the optimal airfoil at $\lambda = 2.5$ (see Fig. 15a).
Overall, at $\lambda \leq 3.5$, the $xt_{opt}/c$ belongs to the range of $xt/c$, which corresponds to the highest sensitivity of $C_P$ to $r_{LE}$. For example,
the optimal airfoil at $\lambda = 2.5$ (i.e., the NACA0024-4.5/3.5) has $xt/c = 35\%$ that fits in the range of $30\% \leq xt/c \leq 40\%$, within which
the impact of $r_{LE}$ is the most significant. This is while the $xt_{opt}/c$ for $\lambda \geq 4.5$ (i.e., $xt/c = 22.5\%$) does not belong to such a range of
$xt/c$ (i.e., $xt/c \geq 30\%$). In addition, the most noticeable improvement in $C_P$ due to changing the $r_{LE}$ occurs at $\lambda = 2.5$, where the
dynamic stall deeply affects the aerodynamic and power performance of the blade. By increasing $\lambda$ and thus, alleviating or avoiding
the dynamic stall, the aerodynamic loads are less affected by the $r_{LE}$.

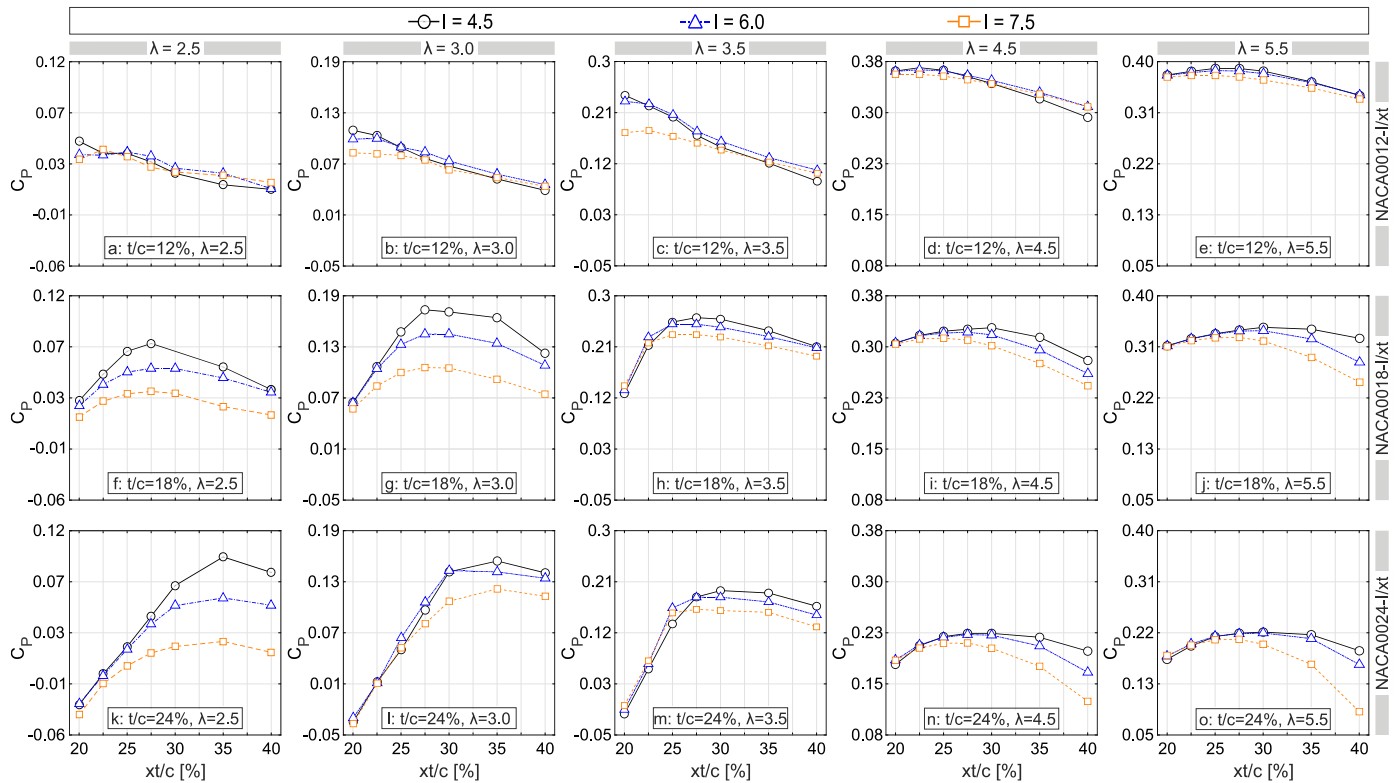

**Figure 14: Comparison of the turbine $C_P$ versus $xt/c$ for different $I$ and $\lambda$.**

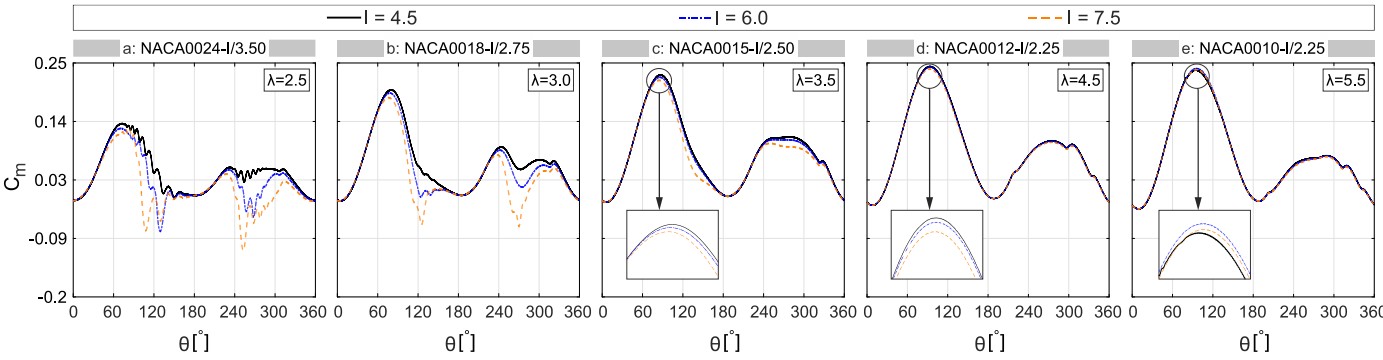

**Figure 15: Impact of changing $r_{LE}$ on the turbine $C_m$ for the combination of $t_{opt}/c$ and $xt_{opt}/c$ at different $\lambda$.**

### 4.1.4  Combined modification of the airfoil shape-defining parameters

The airfoil shape-defining parameters have a coupled impact on turbine performance. Thus, it is of high importance to study the impact of their combined modification on the turbine $C_P$ and $C_T$. Figure 16 shows the variation of $C_P$ in $t/c - xt/c$ space for different $I$ and $\lambda$. Except for $\lambda = 5.5$, where the combination of $t_{opt}/c$ and $xt_{opt}/c$ is achieved by the moderate $I = 6.0$, the $C_{p,max}$ corresponds to the smallest $I = 4.5$ for $\lambda \leq 4.5$.

For $\lambda = 2.5$ and $I = 4.5$, the global optimum occurs by a set of high $t/c$ and $xt/c$ (i.e., NACA0024-4.5/3.50). The combination of $t_{opt}/c$ and $xt_{opt}/c$ values remains invariant for $I = 6.0$; however, the region of maximum $C_p$ shows lower values of $C_p$. For $I = 7.5$, the optimal airfoil changes to a thin airfoil with low $xt/c$, while experiencing lower $C_P$ compared to those of $I = 4.5$ and 6.0. The variation of optimal airfoil shape-defining parameters for different $\lambda$ and the resultant airfoils at each $\lambda$ are illustrated in Fig. 17.

At $\lambda = 3.0$, the region of $C_{P,max}$ shows less sensitivity to $I$, shifting between moderate and high values of $t/c$ and $xt/c$ (see Fig. 16d-f). However, similar to that of $\lambda = 2.5$, the overall range of $C_p$ values narrows down with increasing $I$. For $\lambda = 3.5$, the optimum region of $C_P$ remains nearly the same at moderate values of $t/c$ and $xt/c$ for different $I$ (see Fig. 16g-i); while for higher values of $\lambda$

≥ 4.5, it stays approximately independent of *I*, shifting marginally between low values of *t/c* and *xt/c* (see Fig. 16j-o). This implies
that, by increasing *λ*, the optimum region of turbine $C_P$ is less sensitive to *I*. Overall, by increasing *λ* the local region of optimal
airfoil shape-defining parameters changes from the combination of high values of *t/c* and *xt/c* for *λ* = 2.5 to moderate *t/c* and *xt/c*
for *λ* = 3.0 and 3.5, and low values of *t/c* and *xt/c* for *λ* ≥ 4.5.

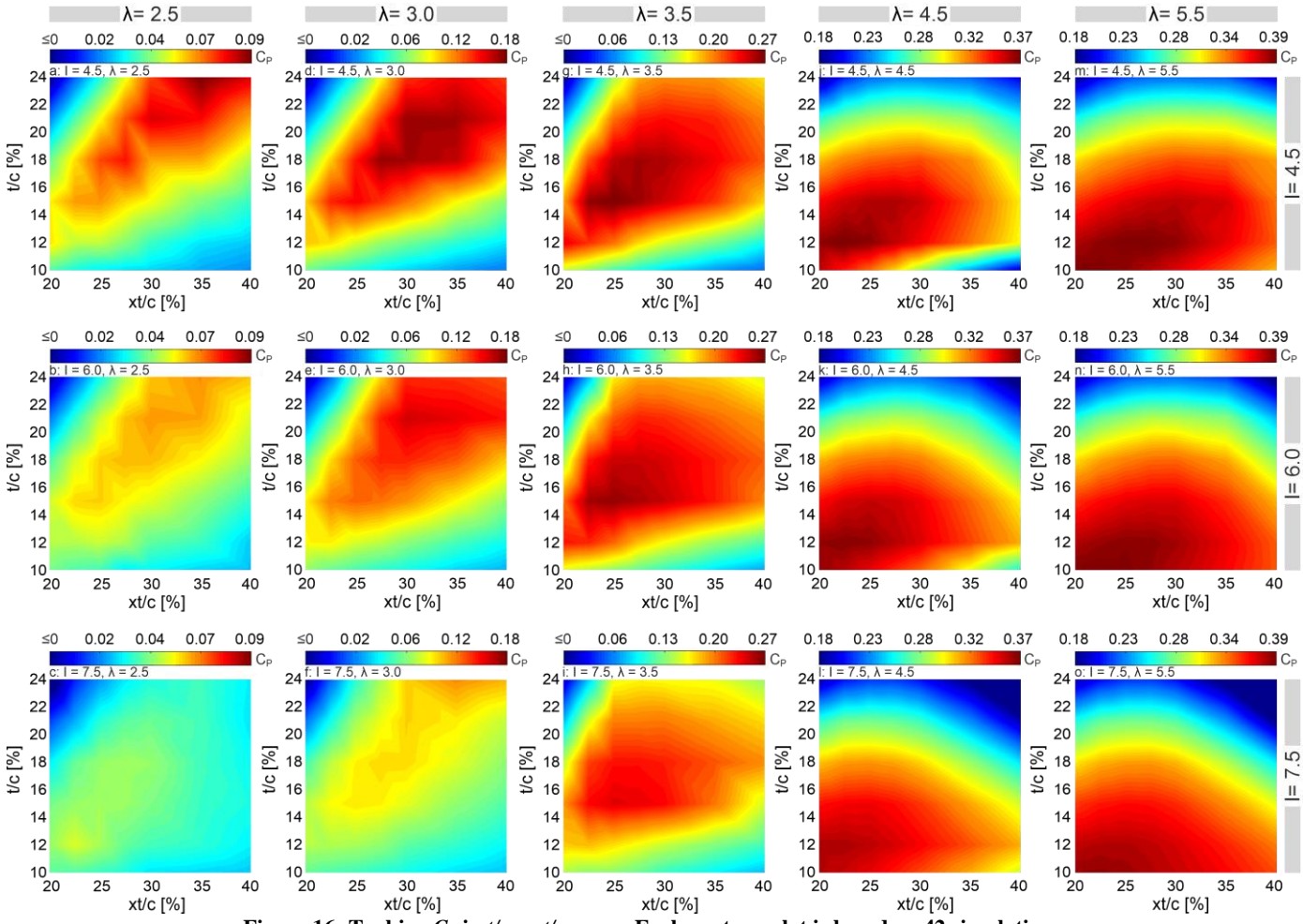

**Figure 16: Turbine $C_P$ in *t/c – xt/c* space. Each contour plot is based on 42 simulations.**
The results highlight that, in designing morphing blades, single-parameter studies will not provide the overall picture and could
lead to unreliable results. The contour plots give a conceptual view of the optimal regions in terms of the airfoil shape-defining
parameters, with which the resultant airfoils have their most efficient performance; and also, the inefficient regions of the turbine
$C_P$, which must be avoided.
Figure 18 shows the turbine $C_T$ in *t/c – xt/c* space. It is interesting to observe that for low *λ* ≤ 3.5 there is no coincidence between
the optimal regions of $C_T$ and $C_P$ contours; while for *λ* ≥ 4.5, these two regions overlap. By increasing *λ*, the optimal region extends
marginally towards higher *t/c* and *xt/c*, while also experiencing higher values of $C_T$. The noncongruent region of $C_{p,max}$ and $C_{T,max}$
at low values of *λ* is different from what is observed in the case of HAWTs. That is, the maximum power output of a HAWT occurs
where the highest thrust load is exerted by the turbine blade on the flow. This led to a correlation between the regions of maximum
$C_P$ and $C_T$. In contrast, the results of the present study show that for VAWTs, the same phenomenon only occurs at high values of
*λ* ≥ 4.5, where the turbine goes into non-dynamic stall regimes with more limited variations of *α*. Therefore, when designing
morphing blades for VAWTs, the $C_T$ values corresponding to high values of *λ* are of more importance compared to those of lower
*λ*, where dynamic stall is expected to occur.

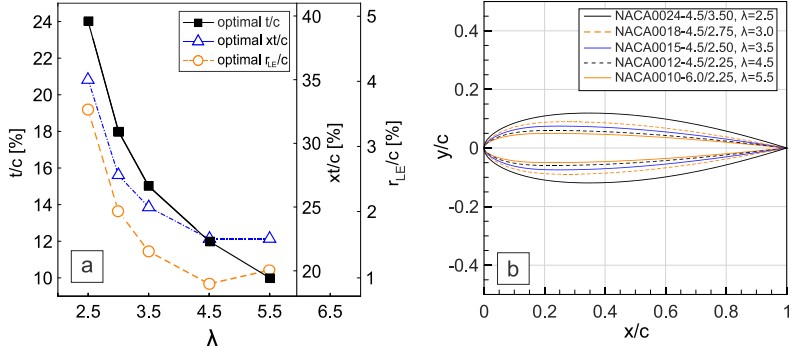

 **Figure 17. (a): variations of the optimal airfoil shape-defining parameters and (b): optimal airfoil shapes at different λ.**


**Figure 18: Turbine $C_T$ in $t/c – xt/c$ space. Each contour plot is based on 42 simulations.**
**4.2 Towards a morphing blade**

This section provides an overview of the turbine power gain due to different morphed-airfoil scenarios, namely a fixed optimal
airfoil for each λ (scenario 1), as already discussed in Sect. 1; and an optimal airfoil for each $d\theta$ (scenario 2), as discussed in the
following Section. Figure 19 shows the variation of $t/c$ and $xt/c$ versus azimuth for scenario 2. Figure 20 shows the corresponding
$C_{m,max}$ for each scenario in $\lambda – \theta$ space. Note that scenario 2 is divided into three cases, namely cases A, B and C. In cases A and
B, the $t_{opt}/c$ and $xt_{opt}/c$ of the already identified optimal shapes for each λ are kept fixed and distributions of $xt/c$ and $t/c$ versus $\theta$,
corresponding to $C_{m,max}$, are extracted, respectively. In case C, the combination of $t_{opt}/c$ and $xt_{opt}/c$, corresponding to $C_{m,max}$ at each
$d\theta$ is selected and kept fixed, and distributions of $xt_{opt}/c$ (i.e., case C1) and $t_{opt}/c$ (i.e., case C2) versus azimuth are extracted,
respectively. Note that $I_{opt} = 4.5$ remains invariant for $\lambda \leq 4.5$ and changes to $I_{opt} = 6.0$ only at $\lambda = 5.5$. For the sake of clarity and
analysis, $I_{opt} = 4.5$ is assumed to be constant throughout the studied range of $\lambda$, introducing the NACA0012-4.5/2.50 as the optimal
airfoil at $\lambda = 5.5$. The relative difference between the $C_{P,max}$ values for optimal airfoils with $I = 4.5$ and 6.0 at $\lambda = 5.5$ is -0.0013.

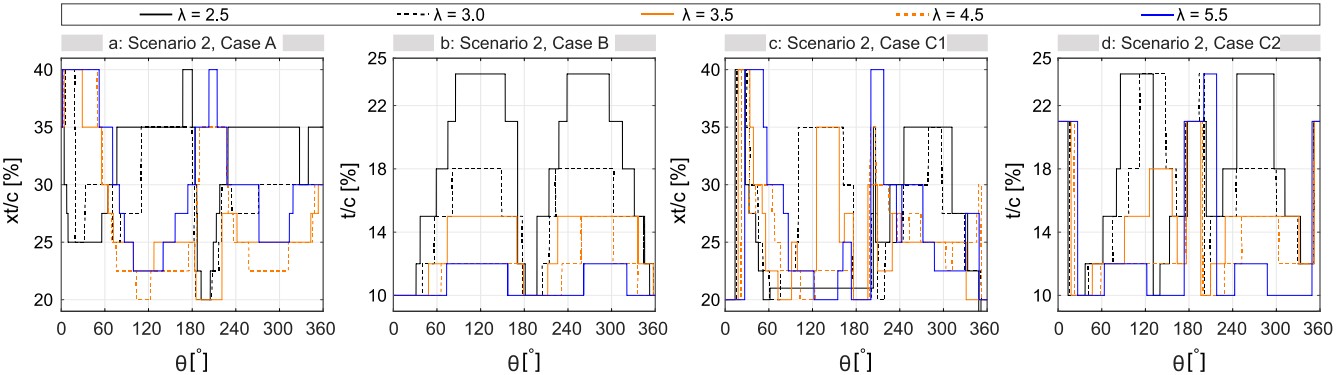

**Figure 19: Changing (a) *xt/c* and (b) *t/c* versuss azimuth for fixed *$t_{opt}/c$* and *$xt_{opt}/c$* corresponding to each *λ*; changing (c) *$xt_{opt}/c$* and (d)**
***$t_{opt}/c$* vs azimuth for fixed *$t_{opt}/c$* and *$xt_{opt}/c$* corresponding to each *dθ*.**

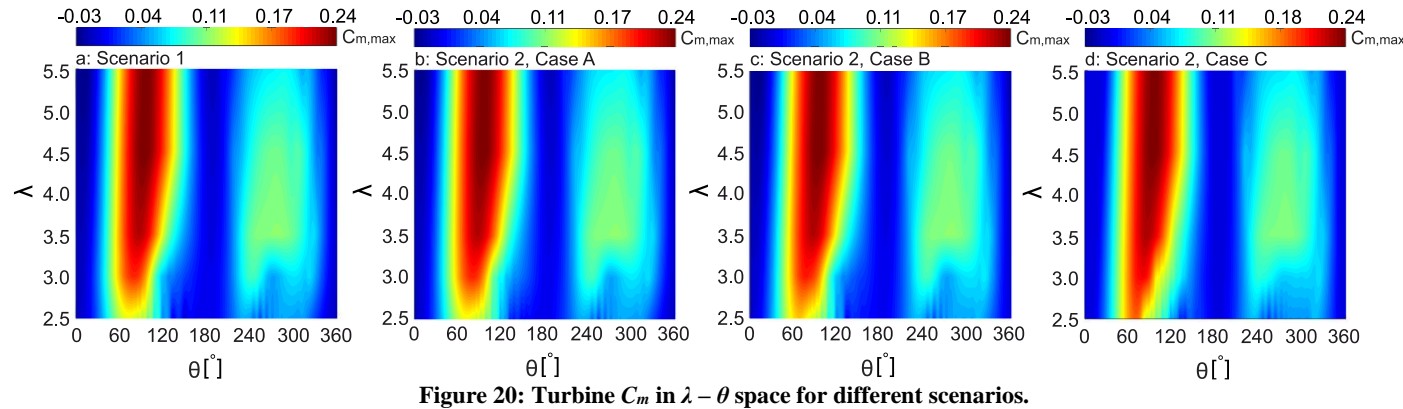

**Figure 20: Turbine *$C_m$* in *λ – θ* space for different scenarios.**

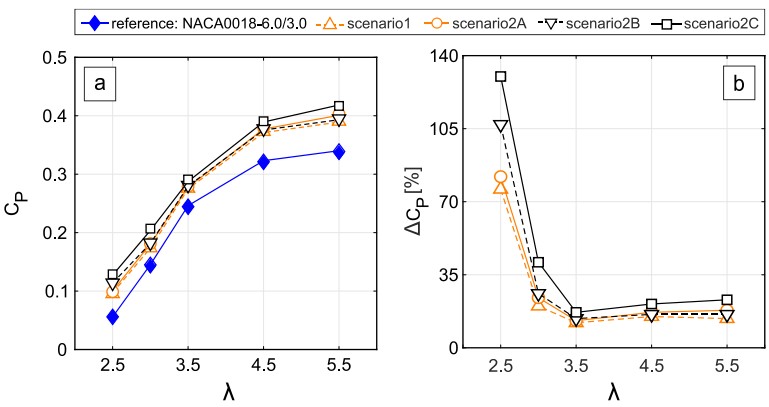

**Figure 21: Variations of (a): turbine *$C_P$* and (b): power gain due to different scenarios for a morphed blade at different *λ*.**
Figures 19a and 20b show the results for scenario 2, case A. Note that the results are based on individual simulations for the studied
airfoil shapes, and correspond to the $xt/c$ with the highest value of $C_m$ at each $d\theta$. It can be observed that $xt/c$ shows almost the
same level of sensitivity to $\theta$ for different $\lambda$ (see Fig. 18 a). Fig. 20b shows the overall view of the $C_{m,max}$ as the blade airfoil is
morphed for different azimuthal position at each $\lambda$. Obviously, the maximum torque is obtained around $\theta = 90°$ for different $\lambda$. The
higher torque generated in the upwind quartile is due to the unperturbed upstream wind profile, while the less pronounced $C_{m,max}$
in the downwind quartile is due to the lower wind velocity and blade-wake interaction.

For scenario 2, case B, the observed trend for $t/c - \theta$ is quite similar for different $\lambda$, except for a noticeable difference; that is, the higher $\lambda$ is, the less sensitive the variation of $t/c$ to $\theta$ is. By increasing $\lambda$, and thus, decreasing the $xt_{opt}/c$, thinner airfoils outperform the thicker ones (see Figs. 19b). The turbine $C_{m,max}$ in $\lambda - \theta$ space shows negligible changes compared to that of scenario 2A (see Figs. 20c). The observations for scenario 2C1 and 2C2 are almost similar to those of cases A and B, respectively. However, there are some narrow ranges of $\theta$ at the beginning, middle, and end of the turbine rotation disk, where noticeable differences exist. The resulting $C_{m,max}$ in $\lambda - \theta$ space differs slightly from the other scenarios (see Fig. 20d).

Figure 21 shows the turbine $C_P$ and the power gain due to the morphed airfoils and the reference case for the studied range of $\lambda$. The highest average improvement in the turbine $C_P$ is due to scenario 2C (i.e., fixed $t_{opt}/c$ and $xt_{opt}/c$, corresponding to the $C_{m,max}$ at each $d\theta$). By increasing $\lambda$ from 2.5 to 3.5, the power gain significantly decreases. Nevertheless, for $\lambda \geq 4.5$ it marginally increases. The more pronounced $\Delta C_P$ at low $\lambda$ is mainly because of alleviating the dynamic stall characteristics due to the morphed airfoil. The averaged improvement in $C_P$ due to scenarios 1, 2A, 2B, and 2C $(\overline{\Delta C_P})$ over the studied range of $\lambda$ is 0.04, 0.045, 0.047, and 0.06, respectively.

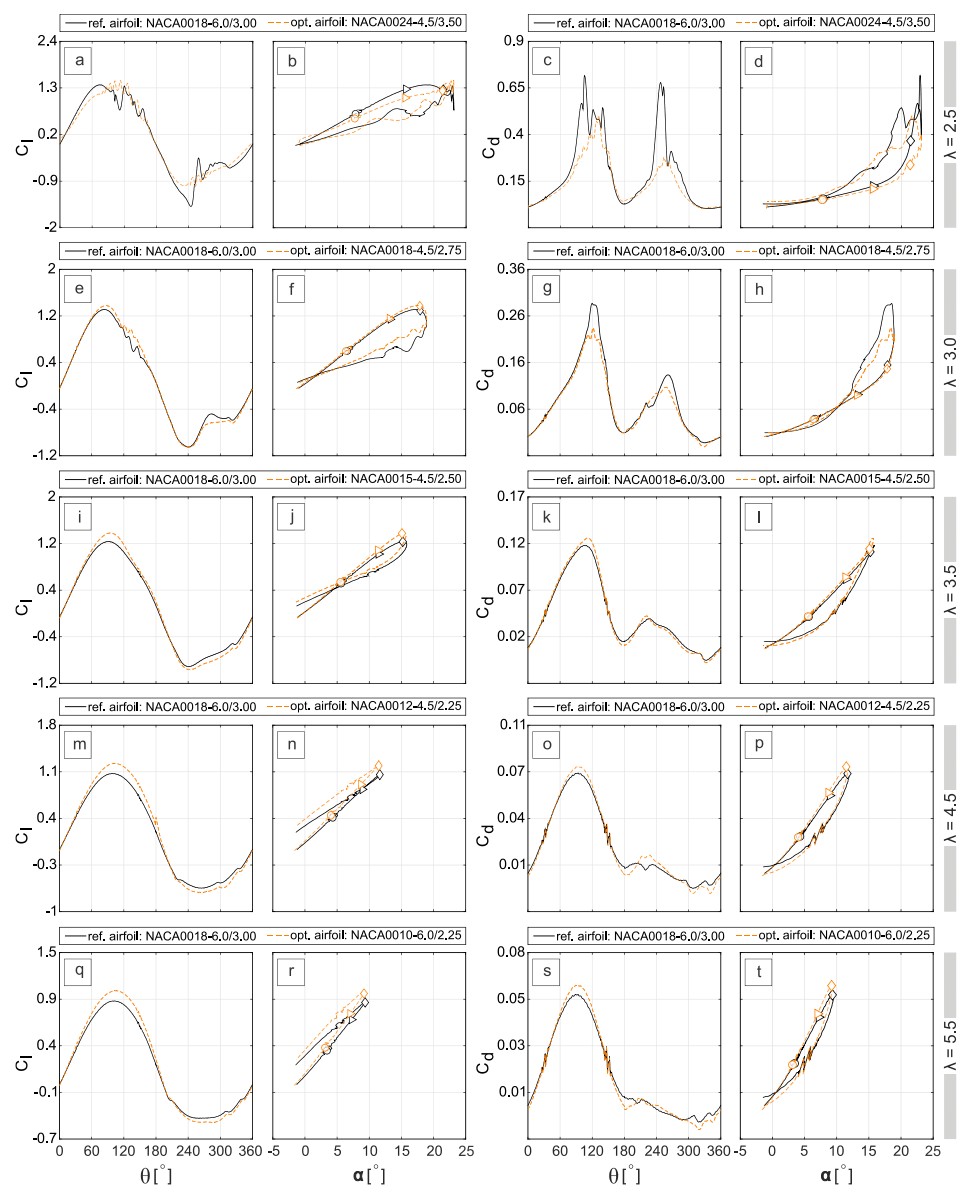

**415**

**416**     **Figure 22: $C_l$ and $C_d$ versus $\theta$ and $\alpha$ for the reference and optimal airfoils at different $\lambda$ ($\bigcirc$: $\theta = 60°$; $\triangleright$: $\theta = 120°$; $\diamondsuit$: $\theta = 180°$).**

**417**     **4.3 Aerodynamic analysis of the morphed airfoils**

**418**     Figure 22 gives a comparison of the turbine aerodynamic loads (namely, $C_l$ and $C_d$) versus $\theta$ and $\alpha$ for the reference and modified

**419**     airfoils. The results correspond to scenario 1, where an optimal airfoil is identified for each $\lambda$. In general, the optimal airfoils have

**420**     higher $C_{l,max}$ compared to that of the reference case. For $\lambda = 2.5$, the optimal airfoil shows an obvious reduction in drag jump both

**421**     in upwind and downwind quartiles and reduced post-stall fluctuation. These are the reflections of the significantly-alleviated

**422**     dynamic stall. Table 5 gives the $C_{l,max}$ and $C_{d,max}$ values for the reference and optimal airfoils at different $\lambda$. It can be seen that for

**423**     $\lambda = 3.0$ and 3.5, where the turbine goes into a lighter-dynamic stall regime, the optimal airfoil shows higher $C_{l,max}$ with less severe

**424**     post-stall fluctuation and lower $C_{d,max}$ with less substantial drag jump. For $\lambda \geq 4.5$ (i.e., non-dynamic stall regime), although the

**425**     modified airfoils show higher values for both the $C_{l,max}$ and $C_{d,max}$, the increase in $C_{l,max}$ is more dominant than that of the $C_{d,max}$

**426**     (see also Table 5). Figure 23 shows the turbine $C_m$ for the reference and optimal airfoils at each $\lambda$. Other than a reduction for $0° \leq$

**427**     $\theta \leq 80°$ at $\lambda = 2.5$, the turbine $C_m$ is found to improve moderately due to the optimal airfoils at the studied range of $\lambda$, indicating

**428**     higher turbine $C_P$.

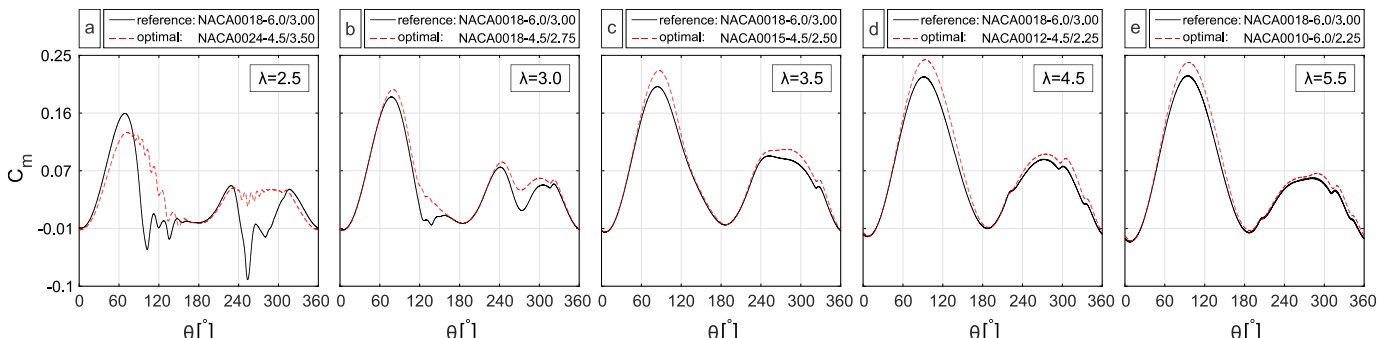

**Figure 23: Turbine $C_m$ for the reference and optimal airfoils at different $\lambda$ (scenario 1).**

**Table 5: Estimated $C_{l,max}$ and $C_{d,,max}$ for the reference and optimal airfoils at different $\lambda$ (scenario1).**

| $\lambda$ | 2.5 | | 3.0 | | 3.5 | | 4.5 | | 5.5 | |
|---|---|---|---|---|---|---|---|---|---|---|
| load coefficient | $C_{l,max}$ | $C_{d,max}$ | $C_{l,max}$ | $C_{d,max}$ | $C_{l,max}$ | $C_{d,max}$ | $C_{l,max}$ | $C_{d,max}$ | $C_{l,max}$ | $C_{d,max}$ |
| reference | 1.37 | 0.716 | 1.31 | 0.29 | 1.23 | 0.118 | 1.07 | 0.076 | 0.93 | 0.064 |
| modified | 1.47 | 0.503 | 1.38 | 0.24 | 1.38 | 0.126 | 1.23 | 0.082 | 1.05 | 0.059 |
| Difference [%] | +7 | -29.8 | +5 | -17 | +12 | +6.7 | +15 | +8 | +13 | +8.5 |

## 5. Discussion

The present work includes a wide range of $\lambda$, where the turbine goes into different operational regimes of light-, deep-, and non-dynamic stall regimes. The aim of the analysis is to highlight the power gain of VAWTs due to different morphed-airfoil scenarios. The results prove the usefulness of the morphing technique to improve the power performance of VAWTs as the main objective of this work. Also, the structural strength of the blade could be another important objective that must be considered while designing morphing blades for VAWTs. It is found that this objective is also satisfied, and the blade structural limitations are met. This is due to the fact that the morphed airfoil changes from a thin one for the highest $\lambda$, corresponding to low wind speeds and aerodynamic loads, to a more robust thick airfoil for the lowest $\lambda$, where the lack of strength and stiffness can cause blade failure, and thus, the blade needs to withstand the aerodynamic loads and to avoid the resultant deflections. However, the maximum and minimum morphing ranges for the airfoil shape-defining parameters might be limited due to manufacturing process. Another technical challenge of utilizing morphing blade for VAWTs is the fatigue failure of the blade due to continuous shape changing. Therefore, an analysis of stresses and fatigue is of high importance to determine the effects of morphing technique on the lifetime of the smart rotor. In addition, technical considerations related to the complexity of the electromechanical actuators for the morphing blade must be taken into account. The required actuators need to be chosen such that they can meet the displacement requirements at the given response times and rotational speeds in Table 6, which might be unfeasible for very small values of d$\theta$. However, extracting the optimal airfoils corresponding to higher values of d$\theta$ (e.g., d$\theta$ = 30°, 45°, and 90°) could result in much higher values of response time and thus, makes it technically possible to adapt the shape changes with azimuthal position. It is of particular importance to consider the cost factor and also to estimate the contribution of morphing blade in annual energy production of the wind turbine for an annual average wind speed, i.e., the difference between the power required to drive the actuators and the resulting turbine power gain.

**Table 6: Actuator response time for the blade to morph at $\lambda$ = 2.5, 3.0, 3.5, 4.5 and 5.5.**

| $\lambda$ | $\Omega$ (rad/sec) | $\Omega$ (deg/sec) | RPS | Response time (ms) |
|---|---|---|---|---|
| 2.5 | 46.5 | 2664 | 7.4 | 0.37 |
| 3.0 | 55.8 | 3197 | 8.8 | 0.31 |
| 3.5 | 65.1 | 3730 | 10.4 | 0.27 |
| 4.5 | 83.7 | 4795 | 13.3 | 0.21 |

| 5.5 | 93 | 5328 | 14.8 | 0.19 |
| --- | --- | --- | --- | --- |

*Note: RPS (revolution per second); ms (millisecond)*

## 6. Limitations

### 6.1 Geometrical parameters

The symmetric modified NACA 4-digit airfoil series is chosen as a basis for the studied airfoils. The airfoils are generated by changing the three main defining parameters, i.e., $t/c$, $xt/c$ and $r_{LE}$. However, it is suggested to continue this work for the rest of the parameters, such as camber and its position along the chord, which describe the airfoil asymmetry and have the potential to morph. The number of blades ($n$) and solidity ($\sigma$) are another two important parameters that would also impact the turbine performance. Some attempts have been made to study the impact of these parameters on turbine performance (Rezaeiha et al., 2018a; Subramanian et al., 2017). For example, it was shown that for different $\lambda$, at a given $Re_c$ the variations of $\alpha$ are almost independent of $n$. In addition, increasing solidity decreases the variations of $\alpha$ at different $\lambda$ (Rezaeiha et al., 2018a). Therefore, based on the results presented in sect. 4.2, it is expected that for 2-, 3- and 4-bladed VAWTs, the airfoil shape-defining parameters show the same level of sensitivity to $\theta$; and for higher $\sigma$, the airfoil parameters show less pronounced sensitivity to $\theta$. However, due to high computational costs, the focus of this work as the first step in designing smart rotors, is confined to investigating the impact of airfoil parameters for a single-blade turbine with a fixed solidity. In addition, due to the large number of simulations in this work, the location of the blade-spoke connection is considered fixed at $c/2$. Nonetheless, for real application scenarios, dedicated investigations are required to study the sensitivity of the optimal regions for the airfoil shape-defining parameters to the number of blades, solidity, and the blade/spoke connection point.

### 6.2 Unsteady aerodynamics

The present study is performed based on a quasi-static assumption where the optimal airfoils at each $d\theta$ are selected from individual simulations for the studied airfoil shapes. Therefore, the effect of the varying unsteady change in bound circulation due to the morphing blade has been considered negligible, and hence no shed vorticity is assumed as a result of the bound circulation temporal gradient. The presented results, as the first step on the way to the smart rotor design, can be utilized as primary tools for quasi-dynamic simulations, where a more focused analysis on a morphing blade scenario would inevitably have to include the mentioned effect; but in view of the major aims put forward in this work, this scenario is left for future studies.

### 6.3 Operational parameters

The present study is focused on a fixed Reynolds number ($Re$), turbulence intensity ($TI$) and reduced frequency ($K$). In an extensive numerical study by (Rezaeiha et al., 2018b), it was shown that the variations of $\alpha$ and normalized $V_{rel}$ are almost independent of $Re$ and $TI$. Nevertheless, dedicated studies are mandatory to draw definitive conclusions concerning the impact of these parameters on the optimal region of airfoil geometrical parameters.

### 6.4 Modeling approach

In the present study 2D URANS simulations are conducted, representing the midplane of a turbine with a high aspect ratio and negligible 3D tip effects. The 2D simulations are chosen based on our earlier study, where the results from 2D and 2.5D simulations for a VAWT with a given $\lambda$ and $\sigma$ showed negligible differences (<1%) in power and thrust coefficients ($C_P$ and $C_T$) (Rezaeiha et al., 2017a). However, compared with the more computationally expensive approaches such as scale-resolving simulations (SAS)

and hybrid RANS/LES, the URANS approach fails to provide accurate prediction of the turbine power performance under the influence of the dynamic stall characteristics at low $\lambda$ (i.e., formation, growth, bursting/shedding of the LSB, dynamic stall vortex (DSV), and trailing edge vortex (TEV)) (Rezaeiha et al., 2019a).

## 7. Conclusions

Incompressible URANS simulations, previously validated with experiments, are used to study the impact of different morphed-airfoil scenarios on the power and thrust performance of a VAWT. Three main airfoil shape-defining parameter, namely $t/c$, $xt/c$ and $I$, are chosen and modified as functions of $\lambda$ and $\theta$ to determine the optimal airfoils in terms of $C_P$ in a wide range of $\lambda$.

The main conclusions are as follows:

- For each $\lambda$, there exists an optimal airfoil shape, corresponding to the turbine $C_{P,max}$. At the lowest $\lambda = 2.5$, the modified airfoil is defined with $t/c = 24\%$, $xt/c = 35\%$ and $I = 4.5$ (i.e., the NACA0024-4.5/3.5). In comparison to the baseline airfoil (i.e., the NACA0018-6.0/3.0), this airfoil has a smaller leading-edge radius; and a higher maximum thickness, which is found to shift downstream of the default point by 5%.

- By increasing $\lambda$, the combination of $t_{opt}/c$ and $xt_{opt}/c$ changes to lower values; however, it shows less dependency on $r_{LE}$. For $\lambda = 3.0, 3.5, 4.5,$ and $5.5$, the optimal airfoils are the NACA0018-4.5/2.75, NACA0015-4.5/2.50, NACA0012-4.5/2.25 and NACA0010-6.0/2.25, respectively.

- Regarding the modified airfoil as a function of $\theta$, the highest average improvement in the turbine $C_P$ is due to scenario 2C, where the combination of $t_{opt}/c$ and $xt_{opt}/c$, corresponding to the turbine $C_{m,max}$ at each $d\theta$, is selected and kept fixed.

- The improvement in $C_P$ due to modifying blade becomes more pronounced for low values of $\lambda$, where the adverse effects of dynamic stall, i.e., jump in aerodynamic loads and post-stall loads fluctuation, are mitigated by morphed airfoils.

The presented work not only highlights the strong relevance of the gain in turbine $C_P$ to different scenarios for morphing airfoils but also emphasizes the combined changing of the airfoil shape-defining parameters. That is, single-parameter modification will not result in the highest power improvement of VAWTs. Other important considerations, such as changing the rest of the geometrical parameters (e.g., camber and its chordwise position, blade/spoke connection point, number of blades, and solidity), are yet to be determined. Therefore, the present study could be a significant stride towards future studies on designing advanced morphing blades for smart VAWTs.

**Acknowledgement**

The first author acknowledges the support from his home university for the use of supercomputing facilities. The second author is currently a postdoctoral fellow of the Research Foundation – Flanders (FWO) and is grateful for the financial support (project FWO 12ZP520N).

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
