# Peer review of "Towards Smart Blades for Vertical Axis Wind Turbines: Different Airfoil Shapes and Tip Speed Ratios"

_Wind Energy Science, 2022_

## Author Comment (AC1)

**Response letter for submission of a revision to Wind Energy Science**

Daniel Micallef
University of Malta
Department of Environmental Design, Msida, MSD2080, Malta

Date: 03-22-2023

Dear Editor,
We would like to thank you for your time and care in handling our submission. Additionally, we would like to express our gratitude to the reviewers for their time and the valuable and helpful comments.

All comments have been fully taken into account, as explained below.

Thank you in advance for the consideration of the revised manuscript.

Sincerely,

Mohammad Rasoul Tirandaz
Abdolrahim Rezaeiha
Daniel Micallef

**RC #1:**

The paper presents a wide set of numerical simulations of a virtual 1-blade Darrieus turbine using different airfoils. The amount of data presented is relevant and represents the main point of merit of the study.

**ANSWER:**

Thanks a lot for the appreciation of our work and your comments. All your comments have been carefully addressed. Below each comment, our responses are given in 'blue' and the modified/new text are shown in 'red'.

**Comments 1:**

However, the results obtained for a single blade are scarcely representative of those a full turbine, since blade-to-blade interaction is lost, as well as the effect of turbine solidity on flow induction. This is reasonable for a first study but limits the validity of the conclusions. Double-checking the outcomes for at least 2 or 3 configurations in case of a 2- or 3-blade turbine could help understanding the validity of the analysis.

**ANSWER:**

The number of blades and solidity indeed are important factors that should be considered while designing a morphing blade. We need to kindly mention that in an earlier study, Rezaeiha et al. (please refer to Rezaeiha et al., 2018a in the revised manuscript) systematically investigated the impact of solidity ($\sigma$) and number of blades (n) on the performance of 2-, 3- and 4-bladed Darrieus H-type VAWTs. Due to the high computational cost, in this study we have focused on single-blade turbine. Therefore, this 'double-checking exercise' requested by the reviewer has already been done in a previous work. In order to address more completely the reviewer's comment, we have added a new section within which the limitations of the research are expressed (please refer to sect. 6 in the revised manuscript). The following paragraph in the revised manuscript emphasizes the importance of investigating the impact of solidity and number of blades on the identified optimal regions for the airfoil characteristics.

Sect.6, Subsect.6.1: **Geometrical parameters**
…The number of blades (n) and solidity ($\sigma$) are another two important parameters that would also impact the turbine performance. Some attempts have been made to study the impact of these parameters on turbine performance (Rezaeiha et al., 2018c; Subramanian et al., 2017). For example, it was shown that for different $\lambda$, at a given $Re_c$ the variations of $\alpha$ are almost independent of n. In addition, increasing solidity decreases the variations of $\alpha$ at different $\lambda$ (Rezaeiha et al., 2018c). Therefore, based on the results presented in sect.

4.2, it is expected that for 2-, 3- and 4-bladed VAWTs, the airfoil shape-defining parameters show the same level of sensitivity to $\theta$; and for higher $\sigma$, the airfoil parameters show less pronounced sensitivity to $\theta$. However, due to high computational costs, the focus of this work as the first step in designing smart rotors, is confined to investigating the impact of airfoil parameters for a single-blade turbine with a fixed solidity. In addition, due to the large number of simulations in this work, the location of the blade-spoke connection is considered fixed at $c/2$. Nonetheless, for real application scenarios, dedicated investigations are required to study the sensitivity of the optimal regions for the airfoil shape-defining parameters to the number of blades, solidity, and the blade/spoke connection point.

**Comments 2:**

The term "morphing" is used throughout the paper. However, if this reviewer understood correctly, all simulations refer to an individual geometry tested under different conditions. The authors should more carefully alternate the terms "morphing" with "changing" or "modifying" for better transparency.

**ANSWER:**

We appreciate the reviewer's comment. We have made all changes suggested by the reviewer and performed a careful editing work in the revised manuscript.

**Comment 3:**

"Morphing" a blade during the revolution (second scenario considered) means that the flow induction changes at each azimuthal position, thus the behavior of the blade cannot be reconstructed as the sum of different "pieces" coming from different simulations. This part of the study must be re-thought carefully.

**ANSWER:**

Thanks for the insightful comment. We need to kindly mention that as the first step on the way to the smart rotor design, we have assumed that the effect of the varying unsteady change in bound circulation due to the morphing blade is negligible and hence no shed vorticity is assumed as a result of the bound circulation temporal gradient. Indeed, a more focused analysis on a morphing blade scenario would inevitably have to include this effect but in view of the major aims put forward in this work this scenario is left for future studies. To address your comment, we have added a new section on research limitations in the revised manuscript. The following paragraph is added for the sake of clarity and to emphasize the importance of continuation of this work for a quasi-dynamic analysis, where the aforementioned effects are included.

Sect.6, Subsect.6.2: **Unsteady aerodynamics**
The present study is performed based on a quasi-static assumption where the optimal airfoils at each d$\theta$ are selected from individual simulations for the studied airfoil shapes. Therefore, the effect of the varying unsteady change in bound circulation due to the morphing blade has been considered negligible, and hence no shed vorticity is assumed as a result of the bound circulation temporal gradient. The presented results, as the first step on the way to the smart rotor design, can be utilized as primary tools for quasi-dynamic simulations, where a more focused analysis on a morphing blade scenario would inevitably have to include the mentioned effect; but in view of the major aims put forward in this work, this scenario is left for future studies.

**Comment 4:**

Lumped references (like [23-33] or [42-46]) should be avoided. The authors should try to emphasize the contribution of each cited reference.

**ANSWER:**

We greatly appreciate the reviewer's effort. To address your comment, we have modified the corresponding following paragraphs and performed a careful editing work in the revised manuscript.

Sect.1, Subsect.1.1, Par.2: For example, the effects of morphed trailing edge was studied by (Daynes and Weaver, 2012); in another work, morphing twist was found to reduce the fatigue life of turbine blades (Lachenal et al., 2013); in a work by (Macphee and Beyene, 2015) morphing blade pitch was discovered to improve the performance of HAWTs; effects of morphed trailing edge flap on the aerodynamic load control was investigated by (Zhuang et al., 2020).

Sect.1, Subsect.1.1, Par.4: To this date, the performance of VAWTs, which very often use airfoils used in the helicopter industry (Rezaeiha et al., 2020b; Sahebzadeh et al., 2020), has been studied for airfoil parameters such as thickness-to-chord ratio $t/c$ and camber $C$ as proposed in (Song et al., 2020; Mazarbhuiya et al., 2020; Nguyen and Tran, 2015; Jain and Saha, 2020; Bianchini et al., 2015). More recently, a few studies have been conducted to improve VAWTs performance via optimizing the airfoil shape-defining parameters (e. g., maximum thickness $t/c$, chordwise position of maximum thickness $xt/c$, leading edge radius $r_{LE}$, and camber $C$) (Bedon et al., 2016; Ma et al., 2018; Ismail and Vijayaraghavan, 2015). Briefly summarized, these studies reveal that the airfoil shape strongly influences the torque characteristics and pressure distribution of the rotor; the type of stall mechanism; the aerodynamic load coefficients, namely lift and drag coefficients $C_l$ and $C_d$; the self-starting capability; and the power coefficient of VAWTs. However, the majority of these studies, which include a few numbers of test cases, have addressed the impacts of a single parameter and keeping the others fixed. This is while, it has been shown that the airfoil shape-defining parameters have combined impacts on VAWT performance (Tirandaz and Rezaeiha, 2021).

**Comment 5:**
The Reference section is well populated. However, the authors discarded a number of important references on VAWT design and simulation. The following readings are suggested since directly related to the present work:
http://dx.doi.org/10.1016/j.renene.2015.06.048
http://dx.doi.org/10.1016/j.energy.2015.12.111
http://dx.doi.org/10.1016/j.enconman.2014.10.038

**ANSWER:**
We appreciate the reviewer's comment. To address your comment, we have incorporated the suggested references and performed a carful editing work in the revised manuscript.

**Comment 6:**
The choice of a blade-spoke connection at 0.5c makes all the analysis more complicated, since the pitching moment comes to play (see DOI: 10.1115/1.4034940). Moreover, the authors did not mention anywhere in the paper the impact of flow curvature effects (thicker airfoils in particular may result in ineffective virtual ones) – see: http://dx.doi.org/10.1016/j.enconman.2015.09.053. While these two phenomena are directly captured by CFD and are thus included in your analysis, the impact should be mentioned and possibly analyzed to help the reader interpreting the outcomes of the study.

**ANSWER:**
We much appreciate the reviewer's careful review. The location of blade-spoke connection indeed is an important parameter which its variation could influence the results. However, due to high volume of results, in the present study we have focused on a fixed value of c/2 for the location of blade-spoke connection. To address your comments, the following paragraph has been added to the new section on research limitations (i.e., sect. 6) to show the focus of the present study:

Sect.6.1, Subsect. 6.1, Par. 1: …In addition, due to the large number of simulations in this work, the location of the blade-spoke connection is considered fixed at $c/2$. Nonetheless, for real application scenarios, dedicated investigations are required to study the sensitivity of the optimal regions for the airfoil shape-defining parameters to the number of blades, solidity, and the blade/spoke connection point.

The flow curvature indeed is an important determinant of the blade aerodynamic of Darrieus turbine that its proper consideration will yield performance improvements. We need to kindly mention that the parameter which has the greatest impact on flow curvature effects is the ratio of blade chord to turbine radius (please refer to Migliore et al., 1980 in the revised manuscript). As c/R increases, flow curvature effects become significant. However, for low values of c/R, flow curvature has less pronounced effects on the turbine performance. In the present study, due to low value of c/R = 0.12, the effects of this physical phenomenon are considered to be small. To address your comment, the following paragraph is added in the revised manuscript:

Sect.4, Subsect.4.1.1, Par. 4: The effect of flow curvature on aerodynamic loading is another important physical phenomenon to take into account in predicting the performance of VAWTs. Because of the angular velocity of the turbine rotor blades, the relative flow direction continuously varies along the airfoil chord,

and thus, the blades experience curved streamlines. As a result of this, a symmetrical airfoil with zero pitch angle in the circular path of a VAWT rotor behaves as if it's a cambered airfoil with a non-zero pitch angle in a straight flow (Migliore et al., 1980; Rainbird et al., 2015). The flow curvature effects become less pronounced on a curved airfoil (Coiro et al., 2005). In addition, a blade hinge located at 50% chord length significantly alleviates the flow curvature effects. However, among all the parameters, the ratio of blade chord to turbine rotor radius ($c/R$) has the greatest impact on flow curvature effects (Migliore et al., 1980). For low values of $c/R$ (i.e., low solidity), the blade surface pressure distribution shows negligible differences with respect to that of the no-lift condition (Coiro et al., 2005), indicating less pronounced effects of flow curvature on the performance of low-solidity turbines (Coiro et al., 2005; Rainbird et al., 2015). In this study, due to the low value of $c/R = 0.12$ (i.e., low $\sigma$), the contribution of flow curvature effects is considered to be small.

**Comment 7:**

A literature study is referenced for the AoA calculation. However, that study has been recently overcome by a new one that is recommended for consideration by the authors:
https://doi.org/10.1016/j.enconman.2020.113284

**ANSWER:**
Thanks a lot for your comments. To address your comment, we have added the suggested reference and modified the paragraph below:

Sect.4, Par.1: ...However, in a recent study by (Melani et al., 2020) an ad hoc inverse verification procedure was developed to compare the accuracy of three selected methods in calculating the angle of attack from the CFD flow field, including 3-Points, Line Average, and Trajectory approaches.

**Comment 8:**

A discussion is present about the feasibility of a morphing blade system. However, it is strongly recommended to expand this section. Points to be addressed: 1) "time" for the actuators to move the airfoils in case of a variation along the revolution (it seems indeed unfeasible); 2) maximum change in thickness reasonably allowed by a constructive technology; 3) fatigue; 4) energy spent for morphing vs. increase in efficiency.

**ANSWER:**
Thanks a lot for the appreciation of our work and your comments. To address your comment, we have modified the following paragraph and incorporated the table below in the revised manuscript.

Sect.5, Par.2: …However, the maximum and minimum morphing ranges for the airfoil shape-defining parameters might be limited due to manufacturing process. Another technical challenge of utilizing morphing blade for VAWTs is the fatigue failure of the blade due to continuous shape changing. Therefore, an analysis of stresses and fatigue is of high importance to determine the effects of morphing technique on the lifetime of the smart rotor. In addition, technical considerations related to the complexity of the electromechanical actuators for the morphing blade must be taken into account. The required actuators need to be chosen such that they can meet the displacement requirements at the given response times and rotational speeds in Table 6, which might be unfeasible for very small values of d$\theta$. However, extracting the optimal airfoils corresponding to higher values of d$\theta$ (e.g., d$\theta$ = 30°, 45°, and 90°) could result in much higher values of response time and thus, makes it technically possible to adapt the shape changes with azimuthal position. It is of particular importance to consider the cost factor and also to estimate the contribution of morphing blade in annual energy production of the wind turbine for an annual average wind speed, i.e., the difference between the power required to drive the actuators and the resulting turbine power gain.

**Table 1: Actuator response time for the blade to morph at $\lambda$ = 2.5, 3.0, 3.5, 4.5 and 5.5.**

| $\lambda$ | $\Omega$ (rad/sec) | $\Omega$ (deg/sec) | RPS | Response time (ms) |
|---|---|---|---|---|
| 2.5 | 46.5 | 2664 | 7.4 | 0.37 |
| 3.0 | 55.8 | 3197 | 8.8 | 0.31 |
| 3.5 | 65.1 | 3730 | 10.4 | 0.27 |
| 4.5 | 83.7 | 4795 | 13.3 | 0.21 |
| 5.5 | 93 | 5328 | 14.8 | 0.19 |
| Note: RPS (revolution per second); ms (millisecond) | | | | |

**RC #2:**

Introduction
The paper presents the results of significant modelling campaign completed with the aim of finding optimal aerofoil profiles to maximise production from a vertical axis wind turbine. This work in completed in the context of morphing blades, and the papers key stated aim is to 'To pave the road towards smart blades for VAWTs' through providing 'a set of generalizable conclusions from 630 transient simulations … [to] understand the impact of different morphing blade scenarios on the turbine power performance CP as well as the thrust performance CT'.

**ANSWER:**
Thanks a lot for the appreciation of our work and your comments. All your comments have been carefully addressed. Below each comment, our responses are given in 'blue' and the modified/new text are shown in 'red'.

**Comment 1:**
The results presented represent a significant modelling campaign and the discussion of the results is thorough. However, the authors do not seem to draw generalizable conclusions from the work. The simulations are run for a single rotor geometry and whilst the optimum power increase achievable using morphing blades is found for this, single, rotor geometry there does not seem to be general conclusions put forward (or tested). Later, in the discussion section (line 381), it is stated that the main aim of the paper is to "prove the usefulness of the morphing technique to improve the power performance of VAWTs". This aim can be said to have be achieved.

**ANSWER:**
We greatly appreciate the reviewer's effort. The number of blades indeed is an important factor that should be considered while designing a morphing blade. We need to kindly mention that in an earlier study, Rezaeiha et al. (please refer to Rezaeiha et al., 2018c in the revised manuscript) systematically investigated the impact of number of blades (n) on the performance of 2-, 3- and 4-bladed Darrieus H-type VAWTs. It was shown that at a given $Re_c$, for different values of $\lambda$, the variations of $\alpha$ are almost independent of the number of blades. Therefore, based on the results presented in sect. 4.2 in the revised manuscript, it is expected that the airfoil shape-defining parameters show the same level of sensitivity to $\theta$ for 2-, 3- and 4-bladed VAWTs. Due to the high computational cost, in this study we have focused on single-blade turbine. In order to address the reviewer's comment, we have added a new section within which the limitations of the research are expressed (please refer to sect. 6 in the revised manuscript). The following paragraph in the revised manuscript emphasizes the importance of investigating the impact of solidity and number of blades on the identified optimal regions for the airfoil characteristics.

Sect.6, Subsect.6.1: **Geometrical parameters**
…The number of blades (n) and solidity ($\sigma$) are another two important parameters that would also impact the turbine performance. Some attempts have been made to study the impact of these parameters on turbine performance (Rezaeiha et al., 2018c; Subramanian et al., 2017). For example, it was shown that for different $\lambda$, at a given $Re_c$ the variations of $\alpha$ are almost independent of n. In addition, increasing solidity decreases the variations of $\alpha$ at different $\lambda$ (Rezaeiha et al., 2018c). Therefore, based on the results presented in sect. 4.2, it is expected that for 2-, 3- and 4-bladed VAWTs, the airfoil shape-defining parameters show the same level of sensitivity to $\theta$; and for higher $\sigma$, the airfoil parameters show less pronounced sensitivity to $\theta$. However, due to high computational costs, the focus of this work as the first step in designing smart rotors, is confined to investigating the impact of airfoil parameters for a single-blade turbine with a fixed solidity. In addition, due to the large number of simulations in this work, the location of the blade-spoke connection is considered fixed at $c/2$. Nonetheless, for real application scenarios, dedicated investigations are required to study the sensitivity of the optimal regions for the airfoil shape-defining parameters to the number of blades, solidity, and the blade/spoke connection point.

**Comment 2:**

All simulations were completed with a fixed airfoil geometry. This is a valid approach for demonstrating the potential power performance enhancement achievable by scheduling the airfoil section on the rotor tip speed ratio, however the authors also use this simulation data to schedule the airfoil section on the angle of azimuth. This relies on a quasi-static assumption that is not valid as the aerodynamic forces on the blade are dependent on the time history. This limitation should be discussed thoroughly in the paper, and this section of the paper should be significantly altered or removed.

**ANSWER:**

Thanks for the insightful comment. We need to kindly mention that as the first step on the way to the smart rotor design, we have assumed that the effect of the varying unsteady change in bound circulation due to the morphing blade is negligible and hence no shed vorticity is assumed as a result of the bound circulation temporal gradient. Indeed, a more focused analysis on a morphing blade scenario would inevitably have to include this effect but in view of the major aims put forward in this work this scenario is left for future studies. To address your comment, we have added a new section on research limitations in the revised manuscript. The following paragraph is added for the sake of clarity and to emphasize the importance of continuation of this work for a quasi-dynamic analysis, where the aforementioned effects are included.

Sect.6, Subsect.6.2: **Unsteady aerodynamics**

The present study is performed based on a quasi-static assumption where the optimal airfoils at each $d\theta$ are selected from individual simulations for the studied airfoil shapes. Therefore, the effect of the varying unsteady change in bound circulation due to the morphing blade has been considered negligible, and hence no shed vorticity is assumed as a result of the bound circulation temporal gradient. The presented results, as the first step on the way to the smart rotor design, can be utilized as primary tools for quasi-dynamic simulations, where a more focused analysis on a morphing blade scenario would inevitably have to include the mentioned effect; but in view of the major aims put forward in this work, this scenario is left for future studies.

**Comment 3:**

Two objectives are put forward by the paper: "(i) To pave the road towards smart blades for VAWTs, having the capability of adaptation to different operational conditions" and "(ii) To provide a set of generalizable conclusions from 630 transient simulations for 126 identical airfoils, generated with different values of maximum thickness t/c, chordwise position of maximum thickness xt/c, and leading-edge radius index I at 5 different tip speed ratios λ; and thus, understand the impact of different morphing blade scenarios on the turbine power performance CP as well as the thrust performance CT". The first objective is not concrete and appears to be a motivation that is inherently completed through the completion of the second objective and should be understood as a research motivation rather than a direct objective of the paper.

A third objective is mentioned in the discussion section: "to prove the usefulness of the morphing technique to improve the power performance of VAWTs". This object is the one that is most readily met by the paper, and it is advised that this should be mentioned in section 1.2.

**ANSWER:**

Thanks for the insightful comment. To address your comment, the third objective is added to the following paragraph in the revised manuscript.

Sect.1, Subsect.1.2: iii. To prove the usefulness of the morphing technique as a promising tool to improve the power performance of VAWTs.

**Comment 4:**

The simulation methodology is clearly described and the paper provides references to previous, more thorough, descriptions of the simulation methodology and validation. The wording in both the abstract and introduction imply that the validation of the numerical model is presented in the current work which is misleading, it is advised to change the description of the model validation to the past tense.

**ANSWER:**

We appreciate the reviewer's comment. To address your comment, we have modified the following paragraphs as suggested by the reviewer in the revised manuscript.

Abstract: The analysis is based on 630 high-fidelity transient 2D CFD simulations, previously validated with experiments.
Sect.2, Subsect.2.3: …Three experimental studies with different test conditions previously were used to validate the CFD simulations.

**Comment 5:**

The methods by which aerofoils are parametrically generated is clearly defined and easily reproducible.

The discussion of the analyses and assumptions is split into 3 sections: (a) the modelling paradigm, (b) the rotor configuration, (c) the analysis of optimal aerofoil shape based of tip speed ratio, and (c) the analysis of optimal aerofoil shape based on azimuthal position

1. The simulation model has been validated in previous work referenced by the paper and is a valid method of analysing the performance of vertical axis wind turbines. There is, however, no discussions of the limitations of the modelling paradigm or any justification of why the specific modelling paradigm was used. A short paragraph discussing what phenomena are (and are not) effectively modelled by 2D URANS simulations (i.e. tip losses) and why the results presented here are valid should be included.

**ANSWER:**
We appreciate the reviewer's comments. To address your comment, a new section on research limitations has been added in the revised manuscript (please refer to sect. 6 in the revised manuscript). The following paragraphs address the limitations on operational parameters and modeling approach:

Sect.6, Subsect.6.3: **Operational parameters**
The present study is focused on a fixed Reynolds number ($Re$), turbulence intensity ($TI$) and reduced frequency ($k$). In an extensive numerical study by (Rezaeiha et al., 2018b), it was shown that the variations of $\alpha$ and normalized $V_{rel}$ are almost independent of $Re$ and $TI$. Nevertheless, dedicated studies are mandatory to draw definitive conclusions concerning the impact of these parameters on the optimal region of airfoil geometrical parameters.

Sect.6, Subsect.6.4: **Modeling approach**
In the present study 2D URANS simulations are conducted, representing the midplane of a turbine with a high aspect ratio and negligible 3D tip effects. The 2D simulations are chosen based on our earlier study, where the results from 2D and 2.5D simulations for a VAWT with a given $\lambda$ and $\sigma$ showed negligible differences (<1%) in power and thrust coefficients ($C_P$ and $C_T$) (Rezaeiha et al., 2017a). However, compared with the more computationally expensive approaches such as scale-resolving simulations (SAS) and hybrid RANS/LES, the URANS approach fails to provide accurate prediction of the turbine power performance under the influence of the dynamic stall characteristics at low $\lambda$ (i.e., formation, growth, bursting/shedding of the LSB, dynamic stall vortex (DSV), and trailing edge vortex (TEV)) (Rezaeiha et al., 2019a).

**Comment 6:**

2. The paper uses a single bladed configuration and a fixed solidity over the entire study. In terms of the blade number and previous work is cited to evidence the claim that the blade number does not have a significant effect on rotor performance. However, maintaining a constant solidity, the chord length changes significantly with blade number. This directly impacts both the effective Reynolds number, and the non-dimensional frequency of the aerofoil motion which have impact performance, a short discussion of this would be beneficial. Additionally, the use of a single rotor configuration over the study makes it hard to draw generalisable conclusions (a stated aim of the paper). A discussion/ demonstration of how generalisable conclusions can be drawn from the specific simulation campaign should be included, or the limitations of using only a single rotor configuration should be discussed.

**ANSWER:**

The number of blades and solidity indeed are two important factors that must be considered while designing a morphing blade. We need to kindly mention that in an earlier study, Rezaeiha et al. (please refer to Rezaeiha et al., 2018c in the revised manuscript) systematically investigated the impact of number of blades ($n$) and solidity ($\sigma$) on the performance of 2-, 3- and 4-bladed Darrieus H-type VAWTs. It was shown that at a given $Re_c$, for different values of $\lambda$, the variations of $\alpha$ are almost independent of number of blades. It was also found that increasing $\sigma$ decreases the variations of $\alpha$ for different $\lambda$. Therefore, it is expected that the airfoil parameters show the same level of sensitivity to $\theta$ for 2-, 3- and 4-bladed VAWTs, and for higher values of $\sigma$, show less pronounced sensitivity to $\theta$. In addition, for a given solidity $\sigma$ and at low $\lambda$, the smaller number of blades (higher chord length) was found to deliver a higher $K$ and thus a higher $C_P$. However, due to the high computational cost, in this study we have focused on single-blade turbine with a fixed value of solidity. In order to address the reviewer's comment, we have added a new section within which the limitations of the research are expressed (please refer to sect. 6 in the revised manuscript). The following paragraph in the revised manuscript emphasizes the importance of investigating the impact of number of blades and solidity on the identified optimal regions for the airfoil characteristics.

Sect.6, Subsect.6.1: **Geometrical parameters**
…The number of blades ($n$) and solidity ($\sigma$) are another two important parameters that would also impact the turbine performance. Some attempts have been made to study the impact of these parameters on turbine performance (Rezaeiha et al., 2018c; Subramanian et al., 2017). For example, it was shown that for different $\lambda$, at a given $Re_c$ the variations of $\alpha$ are almost independent of $n$. In addition, increasing solidity decreases the variations of $\alpha$ at different $\lambda$ (Rezaeiha et al., 2018c). Therefore, based on the results presented in sect. 4.2, it is expected that for 2-, 3- and 4-bladed VAWTs, the airfoil shape-defining parameters show the same level of sensitivity to $\theta$; and for higher $\sigma$, the airfoil parameters show less pronounced sensitivity to $\theta$. However, due to high computational costs, the focus of this work as the first step in designing smart rotors, is confined to investigating the impact of airfoil parameters for a single-blade turbine with a fixed solidity. In addition, due to the large number of simulations in this work, the location of the blade-spoke connection is considered fixed at $c/2$. Nonetheless, for real application scenarios, dedicated investigations are required to study the sensitivity of the optimal regions for the airfoil shape-defining parameters to the number of blades, solidity, and the blade/spoke connection point.

**Comment 7:**

3. The analysis of the optimal aerofoil shape based on tip speed ratio is valid

4. The analysis of the optimal aerofoil shape based on azimuthal position is based on the extraction, at each azimuthal coordinate, of the aerofoil shape that corresponds to the maximum local power/torque coefficient. The previous analysis of the aerodynamic behaviour of blade discusses the inherently time-dependent nature of the aerodynamics as the blade rotates, however the extraction (and reconstitution) based on azimuthal coordinate is based in a quasi-static assumption that the aerodynamic characteristics at a given azimuthal coordinate are not dependent on the time history, which is not a valid assumption. This limitation is not discussed in the paper and should be.

**ANSWER:**
Thanks for the insightful comment. We need to kindly mention that as the first step on the way to the smart rotor design, we have assumed that the effect of the varying unsteady change in bound circulation due to the morphing blade is negligible and hence no shed vorticity is assumed as a result of the bound circulation temporal gradient. Indeed, a more focused analysis on a morphing blade scenario would inevitably have to include this effect but in view of the major aims put forward in this work this scenario is left for future studies, To address your comment, we have added a new section on research limitations in the revised manuscript. The following paragraph is added for the sake of clarity and to emphasize the importance of continuation of this work for a quasi-dynamic analysis, where the aforementioned effects are included.

Sect.6, Subsect.6.2: **Unsteady aerodynamics**
The present study is performed based on a quasi-static assumption where the optimal airfoils at each d$\theta$ are selected from individual simulations for the studied airfoil shapes. Therefore, the effect of the varying unsteady change in bound circulation due to the morphing blade has been considered negligible, and hence no shed vorticity is assumed as a result of the bound circulation temporal gradient. The presented results, as the first step on the way to the smart rotor design, can be utilized as primary tools for quasi-dynamic simulations, where a more focused analysis on a morphing blade scenario would inevitably have to include

the mentioned effect; but in view of the major aims put forward in this work, this scenario is left for future studies.

**Comment 8:**

The discussion of the sufficiency of the results to underpin the interpretations will be split between (a) the discussion of the λ dependent blade morphing and (b) the θ dependent blade morphing

1. The results from the λ dependent blade morphing are valid and provide ample data to underpin the conclusions drawn. A minor detail is that, during the discussion, there is often reference to plots of the aerofoil lift, drag and skin friction coefficients which are not graphically presented. Referring to the behaviour in terms of the coefficients, rather than plots of the coefficients would make the lack of inclusion of these plots less jarring to the reader. Otherwise including a few example plots to show the general behaviour may be of benefit to the reader.

**ANSWER:**

We appreciate the reviewer's comment. To address your comment, we have incorporated the following figures suggested by the reviewer and modified the corresponding paragraphs in the revised manuscript.

[Figure]

**Figure 1: Spatiotemporal contour plots of $C_f$ along the suction side of the blade during the first-half of the last revolution for the NACA00t-4.5/27.5 at $\lambda = 2.5$. Note that the X-axis is along the chord line and $\theta=113°$ corresponds to the blade's $\alpha_{max}= 23°$.**

[Figure]

**Figure 2: Impact of $t/c$ on the variations of $C_l$ and $C_d$ versus $\theta$ during the first-half of the turbine last revolution for $xt/c = 27.5\%$ with $I = 4.5$ at $\lambda = 2.5$.**

[Figure]

**Figure 3: Spatiotemporal contour plots of $C_f$ along the suction side of the turbine blade during the first-half of the last revolution for the NACA0012-4.5/xt at $\lambda = 5.5$. Note that the X-axis is along the chord line and $\theta=113°$ corresponds to the blade's $\alpha_{max}= 23°$.**

**Comment 9:**

2. The results from the θ dependent blade morphing are not be valid, as the scheduling of the aerofoil shape is dependent on a quasi-static assumption that is not valid. Presenting these results under the caveat of this limitation may still interesting to the reader, however this limitation must be discussed in the text.

**ANSWER:**

Thanks for the insightful comment. To address your comment, we have added a new section on research limitations in the revised manuscript. The following paragraph is added for the sake of clarity and to emphasize the importance of continuation of this work for a quasi-dynamic analysis, where the aforementioned effects are included.

Sect.6, Subsect.6.2: **Unsteady aerodynamics**

The present study is performed based on a quasi-static assumption where the optimal airfoils at each $d\theta$ are selected from individual simulations for the studied airfoil shapes. Therefore, the effect of the varying unsteady change in bound circulation due to the morphing blade has been considered negligible, and hence no shed vorticity is assumed as a result of the bound circulation temporal gradient. The presented results, as the first step on the way to the smart rotor design, can be utilized as primary tools for quasi-dynamic simulations, where a more focused analysis on a morphing blade scenario would inevitably have to include the mentioned effect; but in view of the major aims put forward in this work, this scenario is left for future studies.

**Comment 10:**

The original contribution of the paper is clearly stated, however the reviewer is not familiar enough with the smart-blade concept to comment on the completeness of the literature review in the introduction. A more full discussion of the key results from the papers cited in the literature view should be included.

**ANSWER:**

Thanks for the insightful comment. To address your comment, the following paragraph is modified in the revised manuscript.

Sect.1, Subsect.1.1, Par.5: For example, in an experimental study by (Pechlivanoglou et al., 2010), positive flap deflection was found to significantly increase lift force while negative flap deflection results in lift reduction, which is effective in rotor deceleration. A numerical study by (Wolff et al., 2014) has shown that morphing trailing edges, specifically the deflection angles and increasing length of the morphing trailing edge, have significant impact on the lift force and thus the stall characteristics of the blade. In another work by (Minetto and Paraschivoiu, 2020) a deformable trailing edge was discovered to alleviate the dynamic stall characteristics and improve the power output of VAWTs. (Tan and Paraschivoiu, 2017) showed that morphing the blade aileron to have the optimal shape for upwind and downwind quartiles can improve the aerodynamic performance of VAWTs. In addition, in a numerical study, it was found that changing the airfoil shape-defining parameters have a substantial impact on the power performance of VAWT operating in the dynamic stall regime (Tirandaz and Rezaeiha, 2021).

**Comment 11:**

The abstract provides a concise and generally complete summary. Presenting the change in peak power coefficient as well as the average change in the power coefficient may be useful. Additionally, presenting the changes in power coefficient as relative (rather than absolute) values would be preferred.

**ANSWER:**

We appreciate the reviewer's comment. We have modified the following paragraph in the revised manuscript.

Abstract: The results show an average relative improvement of 0.46, and an average increase of nearly 0.06 in $C_P$ for all the values of $\lambda$.

**Comment 12:**

The number of references is appropriate however a greater discussion of the references given in the literature review is encouraged.

**ANSWER:**
We greatly appreciate the reviewer's effort. To address your comment, we have modified the corresponding following paragraphs and performed a careful editing work in the revised manuscript.

Sect.1, Subsect.1.1, Par.2: For example, the effects of morphed trailing edge was studied by (Daynes and Weaver, 2012); in another work, morphing twist was found to reduce the fatigue life of turbine blades (Lachenal et al., 2013); in a work by (Macphee and Beyene, 2015) morphing blade pitch was discovered to improve the performance of HAWTs; effects of morphed trailing edge flap on the aerodynamic load control was investigated by (Zhuang et al., 2020).

Sect.1, Subsect.1.1, Par.4: To this date, the performance of VAWTs, which very often use airfoils used in the helicopter industry (Rezaeiha et al., 2020b; Sahebzadeh et al., 2020), has been studied for airfoil parameters such as thickness-to-chord ratio $t/c$ and camber $C$ as proposed in (Song et al., 2020; Mazarbhuiya et al., 2020; Nguyen and Tran, 2015; Jain and Saha, 2020; Bianchini et al., 2015). More recently, a few studies have been conducted to improve VAWTs performance via optimizing the airfoil shape-defining parameters (e. g., maximum thickness $t/c$, chordwise position of maximum thickness $xt/c$, leading edge radius $r_{LE}$, and camber $C$) (Bedon et al., 2016; Ma et al., 2018; Ismail and Vijayaraghavan, 2015). Briefly summarized, these studies reveal that the airfoil shape strongly influences the torque characteristics and pressure distribution of the rotor; the type of stall mechanism; the aerodynamic load coefficients, namely lift and drag coefficients $C_l$ and $C_d$; the self-starting capability; and the power coefficient of VAWTs. However, the majority of these studies, which include a few numbers of test cases, have addressed the impacts of a single parameter and keeping the others fixed. This is while, it has been shown that the airfoil shape-defining parameters have combined impacts on VAWT performance (Tirandaz and Rezaeiha, 2021).

Sect.1, Subsect.1.1, Par.5: For example, in an experimental study by (Pechlivanoglou et al., 2010), positive flap deflection was found to significantly increase lift force while negative flap deflection results in lift reduction, which is effective in rotor deceleration. A numerical study by (Wolff et al., 2014) has shown that morphing trailing edges, specifically the deflection angles and increasing length of the morphing trailing edge, have significant impact on the lift force and thus the stall characteristics of the blade. In another work by (Minetto and Paraschivoiu, 2020) a deformable trailing edge was discovered to alleviate the dynamic stall characteristics and improve the power output of VAWTs. (Tan and Paraschivoiu, 2017) showed that morphing the blade aileron to have the optimal shape for upwind and downwind quartiles can improve the aerodynamic performance of VAWTs. In addition, in a numerical study, it was found that changing the airfoil shape-defining parameters have a substantial impact on the power performance of VAWT operating in the dynamic stall regime (Tirandaz and Rezaeiha, 2021).

**Comment 13:**
Line 14 page 1: Use the past tense for validated as the validation is not presented in the current paper. Additionally include that the CFD simulations are 2D.

**ANSWER:**
Thanks for the reviewer's comments. We have made all changes suggested by the reviewer in the revised manuscript.

**Comment 14:**
Line 16 page 1: Give the relative improvement in power coefficient rather than absolute value and provide the increase in peak power coefficient as well as the increase in mean power coefficient.

**ANSWER:**
We much appreciate the reviewer's careful review. We have made all changes suggested by the reviewer in the revised manuscript.

**Comment 15-17:**
Line 17 page 1: Wind energy not capitalised in key word list.
Line 25 page 2: When was "of the day"? Date the previous reference in the text to give the reader context.
Line 29 page 2: "In contrast" to what? The operational conditions of aircraft, helicopters, UAVs, MAVs, were not discussed in text so the operational conditions of a wind turbine blade are not being defined in contrast to anything.

**ANSWER:**
Thanks for the insightful comment. We have made all changes suggested by the reviewer and performed a careful editing work in the revised manuscript.

**Comment 18:**
Line 31 page 2: Increasing weight and complexity is an issue for wind turbine blades as well.

**ANSWER:**
We greatly appreciate your comments. To address your comment, the following paragraph in the revised manuscript is modified.

Sect.1, Subsect.1.1, Par.2: … that must be overcome in aerospace applications (e.g., additional flight control system and law to handle the complex and large-scale changes in aerodynamic surfaces at both low-speed and high-speed flight conditions) (Beyene and Peffley, 2007).

**Comment 19-21:**
Line 34 page 2: "Through this quasi-sinusoidal variation of α, it exceeds the static stall angle" Replace with "Through this quasi-sinusoidal variation, α often exceeds the static stall angle".
Line 36 page 2: This is the first reference to the tip speed ratio so λ should be defined here (rather than on line 60.
Line 38 page 2: replace 'sophisticated' with complex.

**ANSWER:**
We much appreciate the reviewer's careful review. We have made all changes suggested by the reviewer in the revised manuscript.

**Comment 22 and 23:**
Line 40 page 2: Give more information about references 42-46: What airfoil parameters were changed? What did these papers find? What did they miss?
Line 41 page 2: Again expand on the references 47-49, What results were found in these studies, and how is this piece of work different?

**ANSWER:**
We greatly appreciate the reviewer's effort. To address your comment, we have modified the following paragraphs and performed a careful editing work in the revised manuscript.

Sect.1, Subsect.1.1, Par.2: For example, the effects of morphed trailing edge was studied by (Daynes and Weaver, 2012); in another work, morphing twist was found to reduce the fatigue life of turbine blades (Lachenal et al., 2013); in a work by (Macphee and Beyene, 2015) morphing blade pitch was discovered to improve the performance of HAWTs; effects of morphed trailing edge flap on the aerodynamic load control was investigated by (Zhuang et al., 2020).

Sect.1, Subsect.1.1, Par.4: To this date, the performance of VAWTs, which very often use airfoils used in the helicopter industry (Rezaeiha et al., 2020b; Sahebzadeh et al., 2020), has been studied for airfoil parameters such as thickness-to-chord ratio $t/c$ and camber $C$ as proposed in (Song et al., 2020; Mazarbhuiya et al., 2020; Nguyen and Tran, 2015; Jain and Saha, 2020; Bianchini et al., 2015). More recently, a few studies have been conducted to improve VAWTs performance via optimizing the airfoil shape-defining parameters (e. g., maximum thickness $t/c$, chordwise position of maximum thickness $xt/c$, leading edge radius $r_{LE}$, and camber $C$) (Bedon et al., 2016; Ma et al., 2018; Ismail and Vijayaraghavan, 2015). Briefly summarized, these studies reveal that the airfoil shape strongly influences the torque characteristics and pressure distribution of the rotor; the type of stall mechanism; the aerodynamic load coefficients, namely lift and drag coefficients $C_l$ and $C_d$; the self-starting capability; and the power coefficient of VAWTs. However, the majority of these studies, which include a few numbers of test cases, have addressed the impacts of a single parameter and keeping the others fixed. This is while, it has been shown that the airfoil shape-defining parameters have combined impacts on VAWT performance (Tirandaz and Rezaeiha, 2021).

**Comment 24-33:**
Line 42 page 2: The connective "In other words" doesn't make sense here

Line 45 page 2: 'resulting in improved aerodynamic and power performance'How is aerodynamic performance different to power performance here?

Line 47 page 2:Past tense "have shown" rather than "show"

Line 58 page 2: '126 identical airfoils' implies the same airfoil section was used, perhaps amend with '126 unique airfoils'.

Line 126 page 6: ", not for all the studied airfoils the same event is observed." Replace with ". This behaviour is not observed for all of the studied airfoils."

Line 138 page 6: "can be due to the following reasoning:" replace with "can be explained by the following:"

Line 144 page 6: Describing the trend as polynomial isn't very accurate language as any line can be represented by a high enough order polynomial (including a polynomial of order 1 which is linear), this description is repeated multiple times through the results section and should be replaced with more exact language, perhaps contrast monotonic to non-monotonic or to a line with a defined maxima. References to polynomial are repeated on: Line 145 page 6, Line 191 page 9, Line 203 page 9, Line 208 page 9, Line 209 page 9, Line 211 page 9, Line 215 page 10, Line 216 page 10.

Line 149 page 7: Replace "Overall" with "mean"

Line 192 page 9: "on the one hand" this connective is unnecessary, can just start the sentence, the same goes for Line 203 page 9 "on the other hand"

Line 199 page 9: "This can also be recognized from the Cl and Cd plots, where by increasing xt/c…" This sentence is maybe better to just remove as you have already describing stall onset occurring earlier and without the Cl/Cd plots this description is jarring for the reader.

ANSWER:
Thanks for the comments. We have made all the suggested changes and performed a careful editing work in the revised manuscript.

Line 205 page 10: It is hard to visualise this without a plot of the skin friction coefficient.

ANSWER:
We appreciate the reviewer's comment. To address your comment, we have incorporated the following figures suggested by the reviewer and modified the corresponding paragraphs in the revised manuscript.

[Figure]

Figure 4: Spatiotemporal contour plots of $C_f$ along the suction side of the turbine blade during the first-half of the last revolution for the NACA0012-4.5/$xt$ at $\lambda = 5.5$. Note that the X-axis is along the chord line and $\theta=113°$ corresponds to the blade's $\alpha_{max}= 23°$.

Line 227 page 10: "either" This word should be removed

Line 228/229 page 11: Don't need to reference the Cf contour plots if it is not plotted. Can replace with "This is because increasing xt/c higher than xtopt/c promotes…"

Line 240 page 11: Can just say 'selected' rather than "selected and optimal"

Line 245 page 11: Reference figure 12 when talking about the moment coefficient

Line 283 page 12: Don't need to say "Nevertheless"

Line 290 page 13: "Morphing the airfoil shape-defining parameters is thought to have a fully coupled impact on the turbine CP" replace with "The airfoil shape-defining parameters have a coupled impact on turbine performance"

ANSWER:

We greatly appreciate the reviewer's effort. We have made all the suggested changes by the reviewer and performed a careful editing work in the revised manuscript.

**Comment 43:**
Line 295/296 page 13: "although the combination of topt/c and xtopt/c remains invariant, the local optimum area is found to morph into thin airfoils with low xt/c." It is unclear if the maximum does remain the same, perhaps have a point on the plot shown the maximum would be useful. Also what is meant by the "local optimum area"? this could be edited for clarity.

**ANSWER:**
Thanks for your comments. To address your comment, we have modified the following paragraph and performed a careful editing work in the revised manuscript:

Sect.4, Subsect.4.1.4, Par.2: …The combination of $t_{opt}/c$ and $xt_{opt}/c$ values remains invariant for $I = 6.0$; however, the region of maximum $C_p$ shows lower values of $C_p$. For $I = 7.5$, the optimal airfoil changes to a thin airfoil with low $xt/c$, while experiencing lower $C_P$ compared to those of $I = 4.5$ and 6.0.

**Comment 44:**
Line 302 page 14: "In other words, the higher λ is, the less dependent the local optimum is on I." This is really saying the same thing twice and the sentence can be removed.

**ANSWER:**
We appreciate the reviewer's comments. We have made all the changes suggested by the reviewer and performed a careful editing work in the revised manuscript.

**Comment 45:**
Figure 13 page 14: This represents some of the key, and most interesting, results from the paper and is a great way to visualise a lot data at once. However:
The colormap could be held constant for set of results at a given tip speed ratio, as this would allow easy comparison between all 3 airfoil design parameters that are being varied
Each surface/colormap is made up of only 42 points, and some of the features, such as the 'staircase' appearing at λ=2.5, I = 6, might be due to the interpolation algorithm used to fill the space. Perhaps contour plots would look better? Or perhaps a different plotting function could be used.

**ANSWER:**
Thanks for the comments. We have made all changes suggested by the reviewer and improved the presentation of Fig. 16 in the revised manuscript.

[Figure]

Figure 5: Turbine $C_P$ in $t/c - xt/c$ space. Each contour plot is based on 42 simulations.

Comment 46-48:

Line 315/316 page 14: "optimal" regions of CP and CT is tricky language as optimal value of CT could have multiple meanings, perhaps better to use "regions of maximum CP and CT"
Line 334 page 16: Presenting the relative difference in CP max values is more relevant than the absolute.
Line 361 page 17: Describing the airfoils as "morphing" implies that a dynamic process is taking place, however they were modelled with a constant shape. It would be clearer to describe them as 'morphed' or 'optimal'.

ANSWER:
Thanks for the comments. We have made all changes in the revised manuscript.

Comment 49:

Line 382 page 18: The structural limitations were not thoroughly discussed in the paper

ANSWER:
Thanks a lot for the appreciation of our work and your comments. To address your comment, we have modified the following paragraph and incorporated the table below in the revised manuscript.

Sect.5, Par.2: However, the maximum and minimum morphing ranges for the airfoil shape-defining parameters might be limited due to manufacturing process. Another technical challenge of utilizing morphing blade for VAWTs is the fatigue failure of the blade due to continuous shape changing. Therefore, an analysis of stresses and fatigue is of high importance to determine the effects of morphing technique on the lifetime of the smart rotor. In addition, technical considerations related to the complexity of the electromechanical actuators for the morphing blade must be taken into account. The required actuators need to be chosen such that they can meet the displacement requirements at the given response times and rotational speeds in Table 6, which might be unfeasible for very small values of $d\theta$. However, extracting the optimal airfoils corresponding to higher values of $d\theta$ (e.g., $d\theta = 30°$, $45°$, and $90°$) could result in much

higher values of response time and thus, makes it technically possible to adapt the shape changes with azimuthal position. It is of particular importance to consider the cost factor and also to estimate the contribution of morphing blade in annual energy production of the wind turbine for an annual average wind speed, i.e., the difference between the power required to drive the actuators and the resulting turbine power gain.

**Table 2: Actuator response time for the blade to morph at $\lambda$ = 2.5, 3.0, 3.5, 4.5 and 5.5.**

| $\lambda$ | $\Omega$ (rad/sec) | $\Omega$ (deg/sec) | RPS | Response time (ms) |
|-----------|--------------------|--------------------|------|--------------------|
| 2.5 | 46.5 | 2664 | 7.4 | 0.37 |
| 3.0 | 55.8 | 3197 | 8.8 | 0.31 |
| 3.5 | 65.1 | 3730 | 10.4 | 0.27 |
| 4.5 | 83.7 | 4795 | 13.3 | 0.21 |
| 5.5 | 93 | 5328 | 14.8 | 0.19 |
| *Note: RPS (revolution per second); ms (millisecond)* | | | | |

==Comment 50-53:==

Line 392 page 19: Previously validated with experiments (reference)
Line 396 page 19: Reference to "morphing airfoil shape" could be replaced with "optimal airfoil shape" for clarity.
Line 405 page 19: "A morphing blade" or "morphing blades"
Line 409 page 19: "the rest of the geometrical parameters" Define what these parameters are.

**ANSWER:**

We greatly appreciate the reviewer's effort. We have made all the suggested changes by the reviewer and performed a careful editing work in the revised manuscript.